# Rainfall Trends and Extremes in Saudi Arabia in Recent Decades

**Mansour Almazroui** 

Center of Excellence for Climate Change Research/Department of Meteorology, King Abdulaziz University, P. O. Box 80208, Jeddah 21589, Saudi Arabia; mansour@kau.edu.sa

**Abstract:** The observed records of recent decades show increased economic damage associated with flash flooding in different regions of Saudi Arabia. An increase in extreme rainfall events may cause severe repercussions for the socio-economic sectors of the country. The present study investigated the observed rainfall trends and associated extremes over Saudi Arabia for the 42-year period of 1978–2019. It measured the contribution of extreme events to the total rainfall and calculated the changes to mean and extreme rainfall events over five different climate regions of Saudi Arabia. Rainfall indices were constructed by estimating the extreme characteristics associated with daily rainfall frequency and intensity. The analysis reveals that the annual rainfall is decreasing (5.89 mm decade$^{-1}$, significant at the 90% level) over Saudi Arabia for the entire analysis period, while it increased in the most recent decade. On a monthly scale, the most significant increase (5.44 mm decade$^{-1}$) is observed in November and the largest decrease (1.20 mm decade$^{-1}$) in January. The frequency of intense rainfall events is increasing for the majority of stations over Saudi Arabia, while the frequency of weak events is decreasing. More extreme rainfall events are occurring in the northwest, east, and southwest regions of Saudi Arabia. A daily rainfall of $\geq 26$ mm is identified as the threshold for an extreme event. It is found that the contribution of extreme events to the total rainfall amount varies from region to region and season to season. The most considerable contribution (up to 56%) is found in the southern region in June. Regionally, significant contribution comes from the coastal region, where extreme events contribute, on average, 47% of the total rainfall each month from October to February, with the largest (53%) in November. For the entire country, extreme rainfall contributes most (52%) in November and least (20%) in July, while contributions from different stations are in the 8–50% range of the total rainfall.

**Keywords:** extreme rainfall events; frequency; extreme rainfall contribution; Saudi Arabia

## 1. Introduction

Extreme rainfall events have severe socio-economic impacts, often resulting in droughts, strong winds, and flash floods, and disrupting the environment, including the human population [1,2]. Consequently, any changes in the frequency and intensity of such events are of significant interest. The Intergovernmental Panel on Climate Change (IPCC) has resolved with a high confidence level that the frequency of extreme rainfall events will increase under the influence of global warming in some parts of the world [3]. Climate change often appears as an increase in the intensity and frequency of extreme weather events. Additionally, the trends of climate variables in the historical period may be used as precursors of a changing climate [4]. Therefore, it is essential for us to complete historical trend assessments, including for extreme events [5]. The inter-annual variability of rainfall largely modulates the extreme events in semi-arid to arid regions such as Saudi Arabia, where such events make up a significant fraction of the total annual rainfall. The spatial distributions of inter-annual rainfall over the country stem from climate variability, which is further associated with droughts and disastrous

floods. Therefore, the assessment of heavy rainfall events for specific return periods is often mandatory for the appropriate design of urban drainage and infrastructure, as well as for long-term planning.

Substantial changes in the frequency of extremes can be associated with small changes in variability rather than any change in the mean [6]. The intensification and more frequent occurrence of extreme rainfall have been discussed in climate change studies, and public awareness of the issue has been raised [7]. However, it has not been investigated comprehensively with observational datasets, especially with data at the daily timescale. An evaluation of the trend in extremes on different timescales is vital to develop the adaptation policies that can address the future occurrence of extreme events for a country like Saudi Arabia. Saudi Arabia lies at the tropical latitudes of 18–32° N and longitudes of 35–55° E, and covers 80% of the Arabian Peninsula [8]. In 2019, the country's population reached 34.22 million, up from 27.56 million in 2010 [9], which is a 24% increase in the total population during the last decade. This has imposed severe stress on the water and agricultural resources of the country. Therefore, an in-depth analysis of historical rainfall extreme records is necessary not only for developing policies linked with future planning for the water and agriculture sectors, but also to preserve national heritage sites, promote the tourism industry, and provide disaster resilience.

The global warming phenomenon has restructured rainfall, resulting in frequent extreme events [10–12]. Several studies have been conducted on the trend of rainfall on different spatial and temporal scales to evaluate the potential impacts of climatic changes and variability [8,13–17]. For example, annual and seasonal rainfall along with related extremes over Egypt and the Nile river basin are analyzed using different Mann–Kendall versions for the period 1948–2010 [18,19]. Saudi Arabia is categorized as a semi-arid to arid climatic zone with a low annual rainfall, high temperatures, limited groundwater reserves, and no perennial rivers [20], where future extremes will likely increase [21]. Gauge-based rainfall and climate model-simulated rainfall over Saudi Arabia are assessed against the 53 observations for the period 1965–2005, which discussed the rainfall climatology and trends [22]. The annual mean rainfall over Saudi Arabia is scanty and displays a large spatial and temporal variability over the entire country. The southern parts of the country receive more rainfall compared to the central and northern regions. Saudi Arabia is strongly affected by both severe flash floods and droughts, and the impact of both have intensified in recent years [23,24]. Saudi Arabia has experienced more extreme flood events in the last decade, especially in the northern, eastern, and southwestern regions. An assumption can be made that these more frequent extreme rainfall events are a consequence of global warming, as simulated by the climate models [25]. Heavy rainfall events over Saudi Arabia commonly occur due to deep convection triggered jointly by Mediterranean storms and Active Red Sea Trough events [26]. These intense rainfall events have severe impacts on lives and livelihoods in the major cities of Saudi Arabia. Several extreme events caused floods in Saudi Arabia in recent years [27], such as the 74 mm event on 25 November 2009, and the 111 mm event on 26 January 2011, which both caused severe flooding in Jeddah [8]. Almazroui et al. [28] demonstrated the impact of climate change on the duration of monthly precipitation, both wet and dry spells, using historical records and regional climate model simulations for the Wadi Al Lith basin in western Saudi Arabia.

The pattern of rainfall and its associated atmospheric circulation over Saudi Arabia displays strong seasonality, as the influence of both tropical and extra-tropical drivers (and the interactions between them) causes extreme rainfall events at different times through the annual cycle. During the dry season, the Arabian heat low is driven by the South Asian monsoon, causing middle and upper tropospheric subsidence over the East Mediterranean region and the Middle East [29–31], resulting in the suppression of rainfall over Saudi Arabia. However, mechanical lifting causes rainfall in the southwestern mountainous region due to the effect of the local topography [32]. During the wet season (October–May) the westerly jet passes over Saudi Arabia and interacts with Red Sea Trough (RST), which advects warm, moist tropical air from the south, enhancing the baroclinicity and thus the rainfall in the region [29,32–34]. Moreover, the El Niño Southern Oscillation (ENSO) significantly impacts the wet season precipitation variability over Saudi Arabia through large changes in the moisture

availability, as well as variability in the westerly jet passing over the region. This results in anomalous conditions [35] and triggers rainfall extremes, which are more frequent during the positive phase of ENSO 4 [2].

In addition to the large-scale circulation, the local surface heating and topography also play a crucial role in determining the spatial distribution and intensity of extreme rainfall over Saudi Arabia. A large portion of the annual total rainfall comes from just a few extreme events, which are very rare over Saudi Arabia. In historical records, the extreme daily precipitation events contribute a large fraction (20–70%) of the total rainfall over the Arabian Peninsula [20]. This is likely to be further increased in the future [36]. However, little is still known about how rainfall extremes in different climatic regions and in different seasons contribute to the total rainfall over Saudi Arabia. The concept of a return period is easily inferred, where stationarity in the rainfall regime can be expected. Thus, it is important to establish whether rainfall records show any indication of trends over time [37]. In this study, the observed data from 25 meteorological stations are used to perform a detailed analysis of the temporal and spatial distribution of rainfall, its trend in the present climate, and the contribution of extreme events to total rainfall for a better understanding of the rainfall variability over Saudi Arabia.

The paper is organized as follows: Section 2 describes the data and methodology; Section 3 describes the main results of the study. The summary and conclusion are discussed in Section 4.

## 2. Data and Methods

### 2.1. Data

The daily total rainfall records for the period 1978–2019 were obtained for 25 meteorological stations in Saudi Arabia from the General Authority of Meteorology and Environmental Protection (GAMEP) (Table 1). The geographic locations of the meteorological stations over Saudi Arabia are shown in Figure 1. The rainfall dataset was passed through the quality control test following Almazroui et al. [38] by performing a constancy check related to station relocations, instrument upgrades, and changes in the surrounding environment that were likely to create heterogeneity in the station dataset. The Climate Hazards Group Infrared Rainfall with Stations (CHIRPS; [39]) and Global Precipitation Climatology Centre (GPCC; [40]) gridded monthly rainfall datasets were used to construct the rainfall climatology.

**Table 1.** Surface station information along with coordinates and elevations, the annual and seasonal rainfall (mm) climatology, and their trends (mm decade$^{-1}$). The superscripts a, b, and c represent trend significance at the 90%, 95%, and 99% confidence level, respectively. Stations are arranged in five climate regions of the country.

| Region | Sr. No. | Station Code | Station Name | Rainfall (mm) | | | Trend (mm Decade$^{-1}$) | | | Lat. (N) | Lon. (E) | Elevation (m) |
|---|---|---|---|---|---|---|---|---|---|---|---|---|
| | | | | Annual | Wet | Dry | Annual | Wet | Dry | | | |
| Northern | 1 | 40375 | Tabuk | 31.4 | 30.3 | 0.7 | −2 [c] | −1 | 0 | 28.38 | 36.60 | 778 |
| | 2 | 40356 | Turaif | 84.9 | 83.6 | 0.3 | −14 [b] | −13 [b] | 0 | 31.68 | 38.73 | 855 |
| | 3 | 40360 | Guriat * | 48.7 | 48.0 | 0.4 | 0 | 0 | 0 | 31.40 | 37.28 | 509 |
| | 4 | 40362 | Rafha | 80.6 | 79.4 | 0.1 | −7 | −7 | 0 | 29.62 | 43.48 | 449 |
| | 5 | 40357 | Arar | 59.7 | 58.4 | 0.1 | −3 | −3 | 0 | 30.90 | 41.13 | 555 |
| | 6 | 40361 | Al-Jouf | 56.1 | 55.0 | 0.8 | 5 | 7 | 0 | 29.78 | 40.10 | 689 |
| | 7 | 40394 | Hail | 97.4 | 96.5 | 0.4 | −30 [a] | −28 [a] | 0 | 27.43 | 41.68 | 1015 |
| Interior | 8 | 40405 | Gassim | 131.3 | 130.2 | 0.2 | −19 [c] | −17 [c] | 0 | 26.30 | 43.77 | 648 |
| | 9 | 40437 | Riyadh * | 105.5 | 104.9 | 0.0 | −14 | −12 | 0 | 24.93 | 46.72 | 614 |
| | 10 | 40373 | Al-Qaisumah | 120.7 | 119.9 | 0.3 | −11 [c] | −8 | 0 | 28.32 | 46.13 | 358 |
| | 11 | 40417 | Dammam | 87.4 | 88.9 | 0.2 | 16 | 9 | 0 | 26.45 | 49.82 | 22 |
| | 12 | 40420 | Al-Ahsa * | 86.5 | 84.6 | 0.7 | −3 | −2 | 0 | 25.30 | 49.48 | 179 |
| | 13 | 40430 | Madinah | 61.0 | 56.3 | 4.3 | −5 | −4 | 0 | 24.55 | 39.70 | 654 |
| Coastal | 14 | 40400 | Al-Wejh | 31.1 | 31.0 | 0.1 | 6 | 7 | 0 | 26.20 | 36.48 | 20 |
| | 15 | 40439 | Yenbo | 33.0 | 32.6 | 0.3 | 4 | 5 | 0 | 24.13 | 38.07 | 8 |
| | 16 | 41024 | Jeddah | 51.2 | 50.2 | 0.9 | 1 | 2 | 0 | 21.70 | 39.18 | 15 |
| | 17 | 41030 | Makkah * | 98.1 | 86.5 | 11.6 | −13 [c] | −8 | 1 | 21.43 | 39.77 | 240 |
| | 18 | 41140 | Gizan | 127.8 | 82.2 | 42.3 | 10 | 0 | 8 | 16.88 | 42.58 | 6 |
| Highland | 19 | 41036 | Taif | 162.6 | 132.0 | 29.1 | −5 | −4 | −1 | 21.48 | 40.55 | 1478 |
| | 20 | 41055 | Al-Baha * | 131.5 | 102.3 | 28.9 | −19 [b] | −11 | −7 [c] | 20.30 | 41.65 | 1672 |
| | 21 | 41114 | Khamis Mushait | 184.6 | 118.7 | 65.6 | −12 | −4 | −7 | 18.30 | 42.80 | 2066 |
| | 22 | 41112 | Abha | 221.2 | 165.3 | 55.7 | −25 [c] | −28 [b] | 4 | 18.23 | 42.65 | 2090 |
| Southern | 23 | 41084 | Bisha | 86.4 | 79.8 | 6.2 | −8 | −6 | −2 | 19.98 | 42.63 | 1167 |
| | 24 | 41128 | Najran | 60.3 | 47.4 | 12.9 | 9 | 7 | 2 | 17.62 | 44.42 | 1214 |
| | 25 | 41136 | Sharorah * | 57.8 | 41.0 | 16.6 | −4 | −3 | −1 | 17.47 | 47.10 | 720 |
| | | | *Country* | *92.5* | *80.5* | *11.4* | *−6 [c]* | *−5* | *0* | | | |

Note: Wet season is October–May, and dry season is June–September. The asterisk (*) with the station name represents data available from 1985, while all the others are from 1978, except for Dammam which starts at 1999.

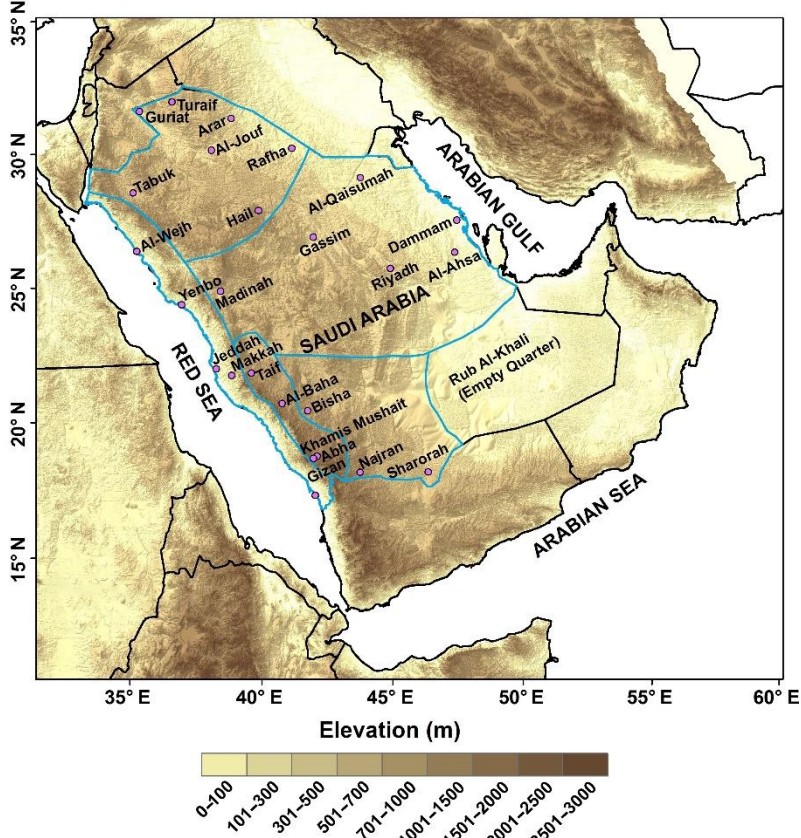

**Figure 1.** Regional map with surface elevation (m) in and around Saudi Arabia. The 25 meteorological station locations across Saudi Arabia (see Table 1) are shown with closed circles. The solid blue lines indicate the five climatic regions—northern (Turaif, Gurait, Arar, Al-Jouf, Rafha, Tabuk, and Hail), coastal (Wejh, Yenbo, Jeddah, Makkah, and Gizan), interior (Al-Qaysumah, Gassim, Dammam, Al-Ahsa, Madinah, Riyadh), highland (Taif, Al-Baha, Abha, and Khamis Mushait), and southern (Bisha, Najran, and Sharorah)—in Saudi Arabia, adapted from Almazroui et al. [41].

## 2.2. Methods

The station-based total rainfall, with the daily temporal resolution, was used to explore the frequency and intensity of rainfall extremes, along with the contribution of extreme events to the total rainfall amount. Extreme rainfall indices were generated using the daily rainfall time series data for the 42-year period 1978–2019. The daily rainfall data is the most reliable way to determine the frequency and trend of extreme rainfall events [42].

The entire time series is divided into two equal periods, and the trend for each half period is then computed along with the decadal trends. The frequency of rainy days was counted at each station for several threshold values, such as ≥1, ≥5, and up to ≥50 mm, with intervals of 5 mm. In this article, an extreme rainfall event is defined using the daily accumulated rainfall. An extreme rainfall event is defined as "An event having at least one day accumulated rainfall equal to or above a specific threshold". The rainfall trends and their significance levels were calculated using the regression equation and F-test, respectively [38]. There is also non-parametric trend analysis, such as the Mann–Kendall (MK) test [43,44] with the MK statistic, and the S test statistic [45] to obtain the slope and significance is also used.

The return period of rainfall occurrence is defined as the time duration expectation over which there are no extreme event occurrences on the average. In practical works, generally, this duration is adapted as 2-year, 5-year, 10-year, 25-year, 50-year, or 100-year, corresponding to 0.50, 0.20, 0.10, 0.04, 0.02, or 0.01 risk levels, respectively. As a method, the extreme event occurrences are assumed to

take place independently from each other, and therefore the probability distribution function (PDF) models provide a scientific basis. There are three distinct approaches, including the (i) annual maxima, (ii) over given threshold extremes, and (iii) threshold selection, such that the number of years is equal to the number of extreme events over the threshold level [46]. Robust assessments can only be obtained from the analysis of the frequency of daily rainfall over a long time-series. The daily total rainfall time-series is analyzed with the help of the gamma distribution. The anticipated annual, wet-season, and dry-season maximum rainfall during the 42-year period is ascertained for different return periods at each station. The return periods are derived from the statistical distribution for every station from the annual extreme time-series; where the trend is obvious in the rainfall regime, a substitute for the concept of return period is suggested, based on the probability of occurrence of an extreme event of specified magnitude, during a period extending into the future during which the observed trend can be assumed to persist.

The threshold for extreme events that contribute to the total rainfall is defined using percentile values. First, the cut-off daily rainfall for the 90th, 95th, and 99th percentiles are obtained. The rainfall amounts at the 90th, 95th, and 99th percentiles are calculated and compared with the thresholds of ≥10 mm up to ≥30 mm with intervals of 1 mm at each station and for the entire country. The amount of percentile-based rainfall equivalent to the threshold-based rainfall defines the extreme event threshold. For example, if the threshold value used to define an extreme event over Saudi Arabia is 26.5 mm day$^{-1}$, the amount of rainfall ≥26.5 mm is equivalent to the amount of rainfall at a specific percentile (e.g., the 95th percentile). Finally, the contribution of extreme events to the total rainfall at different stations is computed in order to understand the extent to which extreme events contribute to the total rainfall over specific climatic regions.

## 3. Results and Discussion

### 3.1. Rainfall Distribution and Climatology

The heterogeneous spatial distribution of rainfall from north to south over Saudi Arabia is shown by the CHIRPS and GPCC data (Figure 2). The wet season extends from October to May and the dry season from June to September. An understanding of the rainfall distribution on different timescales is necessary for climatic and hydrological studies because the amount of rainfall informs projects related to municipal civil works, flood control, agriculture management, and water reservoirs. The climate of Saudi Arabia is influenced by different weather systems due to its wide latitudinal extent. Both tropical and extra-tropical weather systems influence the climate of Saudi Arabia [47]. Seasonal rainfall analysis shows that most rainfall occurs during the spring and winter seasons, while the minimum contribution is from the summer season (Table 1). During the summer season, strong surface heating causes a heat low that is associated with dry conditions over Saudi Arabia. Meanwhile, the Red Sea Trough (RST) extends from south of the Red Sea towards the north over the Eastern Mediterranean. The RST phenomenon favors the development of strong depressions over the central Red Sea, which can lead to heavy rainfall [48]. Furthermore, the remote influence of the ENSO region during its negative phase causes upper-level divergence, along with lower level convergence over the region that favors convective rainfall over the south and southwest of Saudi Arabia [47]. Mechanical lifting by local orography in the southwestern region enhances rainfall on the windward side of the mountain ranges along the Red Sea coast, as seen in Figure 2. The Rub Al-Khali, or Empty Quarter, covering the southeastern region, shows the lowest rainfall totals, with almost no rainfall throughout the year.

Both the CHIRPS and GPCC monthly rainfall datasets show homogeneous spatial rainfall distributions with little quantitative differences. The annual rainfall distribution shows maxima in the central and southwestern parts, with the greatest amounts over 200 mm in the southwestern region (Figure 2a,b). The annual climatology of the CHIRPS dataset shows most rainfall (up to 245 mm) over the southwestern coastal belt and some regions of northeastern Saudi Arabia. Smaller amounts (35 to 70 mm) were observed over the northwestern and southeastern quadrants, while

the central regions received 90 mm to 160 mm during the period 1978–2019. The annual climatology of the GPCC dataset shows a very similar rainfall pattern as in CHIRPS, with maximum rainfall (150–190 mm) occurring over the southwestern coastal regions along with parts of the northeastern region. Less rainfall (approx. 10–70 mm) was measured over the northwestern and southeastern regions, while the central regions acquired values of 60–125 mm. Comparatively, the annual climatology of GPCC shows a somewhat lower rainfall over the country.

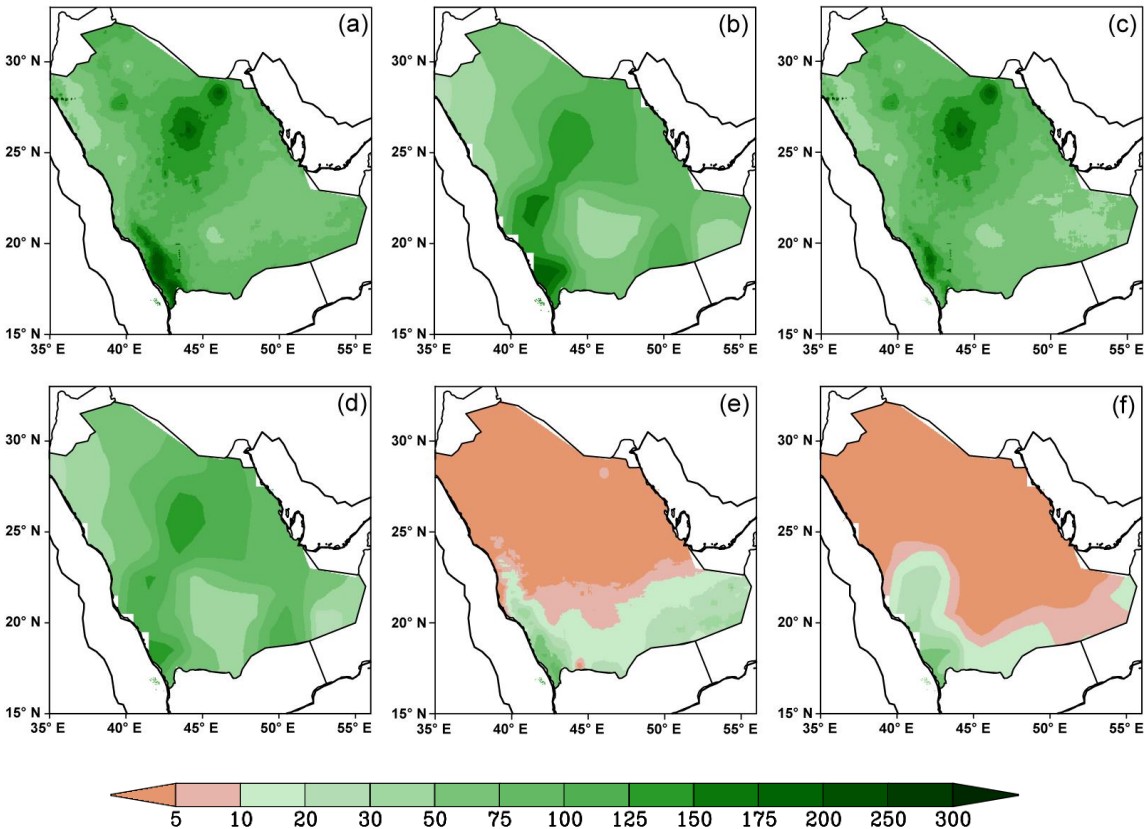

**Figure 2.** Spatial distribution of rainfall (mm) from CHIRPS (**a**,**c**) and GPCC (**b**,**d**) for the whole year (**a**,**b**), wet season (**c**,**d**), and dry season (**e**,**f**) in Saudi Arabia during the period 1981–2019.

The wet season rainfall spatial distribution patterns are influenced through the Mediterranean and central Asian troughs [49,50]. Wet season rainfall distributions show maxima in the central parts of Saudi Arabia, with values of more than 150 mm (Figure 2c,d). Hot and dry conditions prevail during the summer season [51]. However, occasionally, Indian monsoon effects can lead to deep convection over the southwestern highlands. The summer season rainfall distribution shows maxima in southwestern Saudi Arabia, with totals of more than 75 mm (Figure 2e,f). Overall, the CHIRPS data show more specific and higher amounts of rainfall over Saudi Arabia as compared to the GPCC data. Surface rainfall measurements from the meteorological stations across Saudi Arabia show similar values, as tabulated in Table 1.

### 3.2. Rainfall Variability and Trends

The observed mean annual rainfall in Saudi Arabia is 92.5 mm for the period 1978–2019 (Figure 3a, Table 1). The 42-year time series of annual mean rainfall over Saudi Arabia shows an apparent decadal variability (Figure 3a). The time series regression analysis shows a decreasing trend ($-5.89$ mm decade$^{-1}$, significant at the 90% confidence level) for the entire period 1978–2019; it increased at the rate of 21.02 mm decade$^{-1}$ (90% significant level) in the first half (1978–1998), while it had an insignificant slow increase rate (3.86 mm decade$^{-1}$) in the second half (1999–2019). However, the decadal trends vary from the first

to the last decade. In the first decade (1980–1989), the annual rainfall decreased at the rate of −8.52 mm decade$^{-1}$ (insignificant), while in the second decade (1990–1999) it mostly remained above normal and increased at the rate of 32.09 mm decade$^{-1}$ (insignificant). During the first decade, the maximum rainfall occurred in 1981 (approx. 175 mm), whereas the minimum rainfall spike (70 mm) happened in 1984. In the second decade (1990–1999), the maximum rainfall of about 165 mm was observed in 1997. In the third decade (2000–2009), the annual rainfall was mostly below the 42-year mean, and again showed a decreasing trend (−30.70 mm decade$^{-1}$, significant at the 90% confidence level), while the last decade (2010–2019) had an increasing trend of 36.49 mm decade$^{-1}$ (significant at the 90% confidence level). Note that the annual rainfall in 2017 was 70 mm, which is lower than that of the previous year, 2016 (99 mm), and the following year, 2018 (123 mm). However, in 2017 a daily rainfall of 93 and 56 mm fell on February 14 and 17, respectively, in Abha. In Khamis Mushait, a daily rainfall of 51, 63, and 36 mm fell on February 13, 14, and 17, respectively. These suggest that annual rainfall totals come from just a few days and a few stations—i.e., extreme events contribute strongly to the totals.

Similar trend behavior was observed for the wet season rainfall data (Figure 3b). The wet season-observed mean rainfall is 80.5 mm over the full period. The entire period also shows an insignificantly decreasing trend (−4.83 mm decade$^{-1}$), as in the annual rainfall observations. The wet season trends for the first and second half also follows the annual pattern. The wet season rainfall shows an insignificantly decreasing trend (−18.02 mm decade$^{-1}$) during the first decade, while the last decade (2010–2019) shows an insignificantly increasing trend (32.56 mm decade$^{-1}$).

The observed dry season mean total rainfall is 11.4 mm over Saudi Arabia, which accounts for only 12.36% of the annual rainfall during the period 1978–2019 (Figure 3c). Rainfall trends were insignificant in the dry season during all the decadal slots. Overall, the annual and wet season rainfall patterns are quite similar; both depict the rainfall surplus decade of 1990–1999, while the same rainfall deficiency was found in the decade 2000–2009. On the other hand, the dry season rainfall pattern differs slightly from the annual and wet season patterns, since during the dry seasons the largest rainfall deficiency was observed in the decade 1980–1989. The relatively heterogeneous behavior of these trends indicates a large decadal variability in the rainfall over Saudi Arabia. The increasing and decreasing rainfall trends may be due to decadal variations in the large-scale ocean and atmospheric circulation patterns [52].

Because the rainfall data are neither independent nor normally distributed, therefore the slope and significance of trends are also tested using Mann–Kendall test. According to this test, the slopes show increasing and decreasing trends similar to regression analysis, with a slight variation in the magnitude. For example, the annual trends are −5.38, −3.17, 17.41, −32.94, and 55.62 mm decade$^{-1}$ for the entire period, the first decade, the second decade, the third decade, and the last decade, respectively. During the wet season, the trends are −7.64, −27.08, 81.35, −57.07, and 40.64 mm decade$^{-1}$ for the entire period, the first decade, the second decade, the third decade, and the last decade, respectively. In the dry season, the trends are 25.80, 16.45, and −12.42 mm decade$^{-1}$ for the first, second, and third decade, respectively, while closer to the total period and last decade, which are too small. Note that all of them show an insignificant level of confidence in Mann–Kendall test. Different Mann–Kendall versions have almost no change in rainfall and its extremes, but however are very sensitive to temperature extremes [17]. As we are concerned with the precipitation and the Mann–Kendall test shows an almost similar trend to the regression analysis, the trends of the rainfall and extremes are therefore evaluated using the regression method for the rest of the analysis. Note that the rainfall trend for a longer period, such as 30 years or more, is climatologically reasonable for annual and seasonal scales. However, the decadal trends in this study are used to show the variation in different decades, which may not enough for climate study. Therefore, the rest of the study emphasized the use of the 42-year rainfall time series in the trend analysis, which may not sufficient for climate study.

The analysis of climatological rainfall amounts at each station shows that the central and southwestern parts of Saudi Arabia receive the most rainfall, with significantly decreasing trends (Table 1). However, the low rainfall regions such as the eastern, northwestern, and southern

mountainous regions show (insignificant) increasing trends in rainfall over the annual cycle and during the wet season for the period 1978–2019.

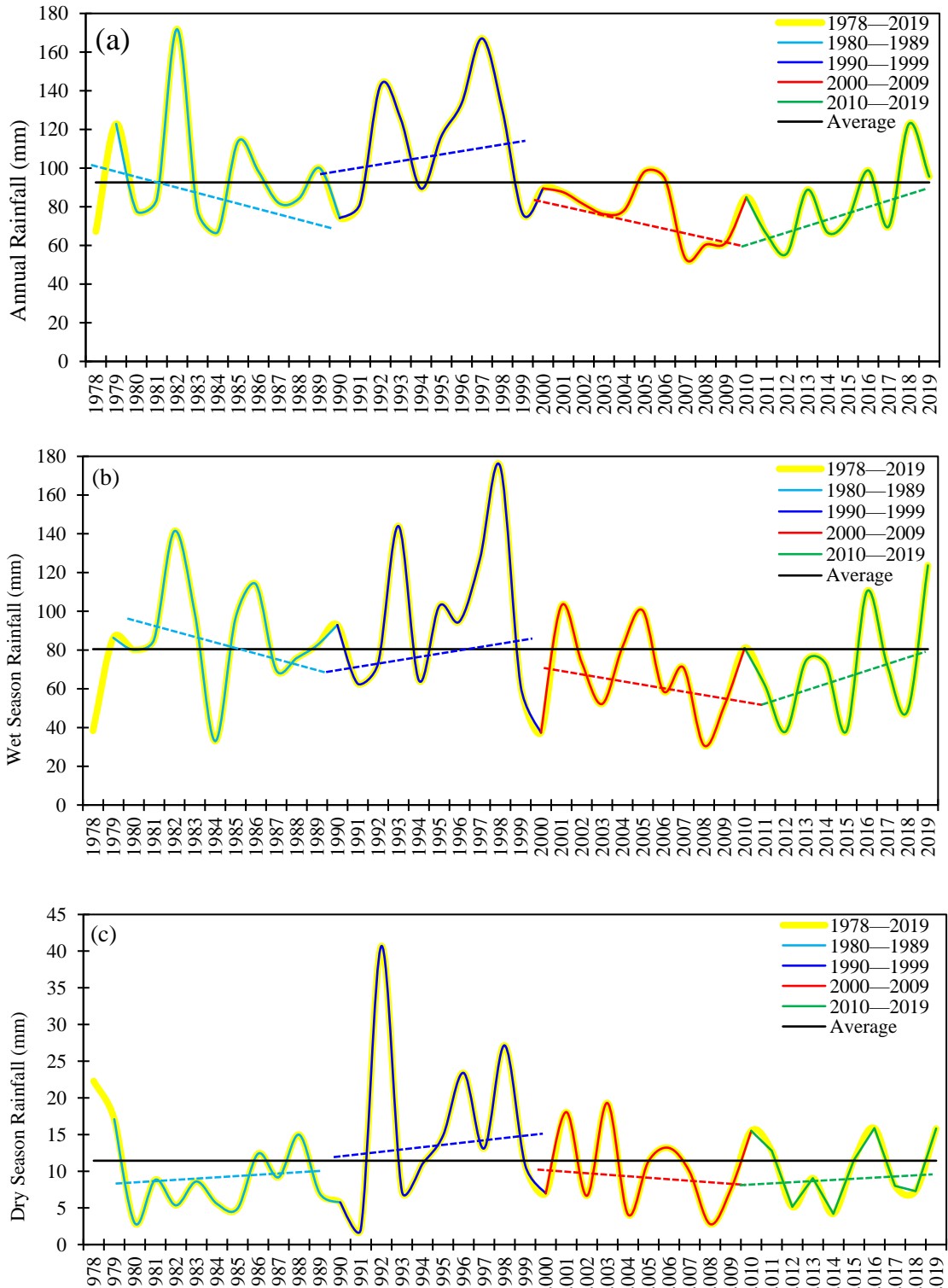

**Figure 3.** Time sequence of the country-averaged rainfall (mm) for (**a**) the whole year, (**b**) wet season, and (**c**) dry season for the period 1978–2019. The decadal trends are also shown at each panel for 1980–1989, 1990–1999, 2000–2009, and 2010–2019. The horizontal line indicates the average rainfall (mm): 92.54 mm for the whole year, 80.47 mm for the wet season, and 11.44 mm for the dry season. Note that the vertical scales are not the same for all the panels.

Figure 4 shows the annual cycle of rainfall, the change with respect to the base period (1981–2010), and the decadal trends over Saudi Arabia. The highest average rainfall (15.83 mm) occurs in April, while the lowest (1.46 mm) occurs in June. The highest positive change was observed during February (with the next highest changes in April and May), while the lowest negative change was observed during June (with the next lowest changes in November and December). There are decreasing trends in the annual rainfall cycle for all months, except for August and November. The highest increase (1.85 mm decade$^{-1}$) was observed in November, and the greatest decrease (−2.83 mm decade$^{-1}$) in March. On the other hand, rainfall changes with respect to the base period 1981 to 2010 show the most positive changes in February, April, May, September, and October, whereas the rainfall amounts were below the mean line in other months of the year. The maximum rainfall change occurred in February, and the minimum is in November. and December.

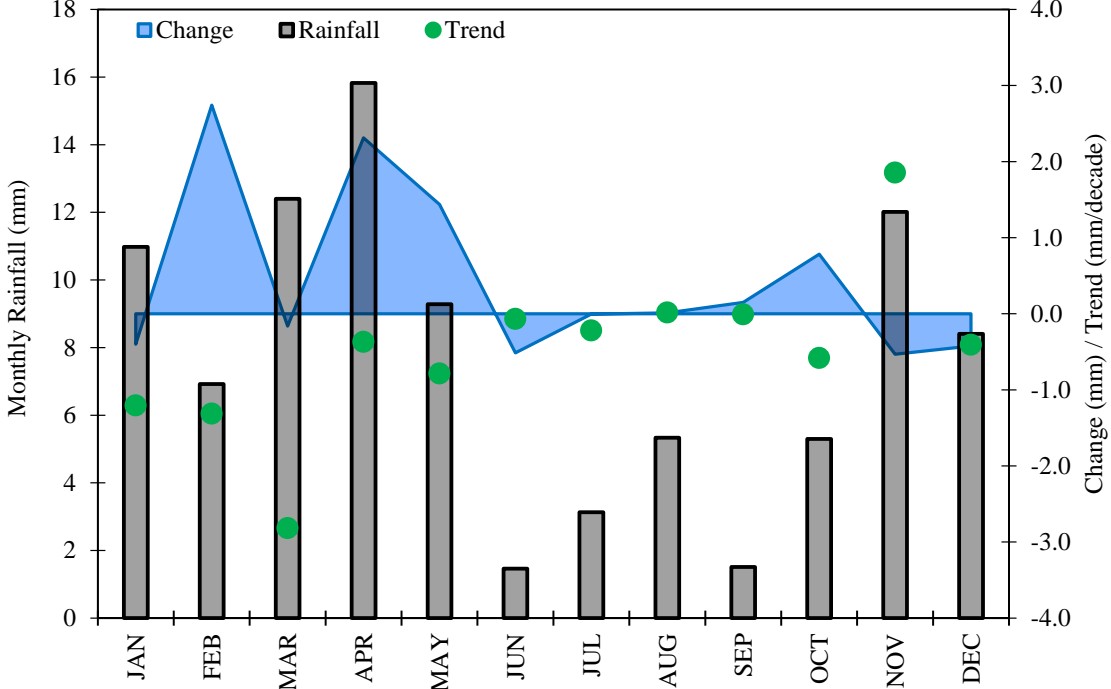

**Figure 4.** Annual cycle of rainfall (mm) along with trends (mm decade$^{-1}$) and change in rainfall (mm) for the period 1978–2019. Note that the change in rainfall is the with respect to base period 1981–2010.

Further, Saudi Arabia can be divided into five climatic regions—i.e., northern, coastal, interior, highland, and southern—as described in [41]. These regions are categorized based on their topographic characteristics and distinct rainfall patterns. The annual cycle of rainfall during some extreme wet (1992, 1997, 2018) and dry (2007, 2008, 2012) years and for the entire period was examined for Saudi Arabia and its five climatic regions (Figure 5). It is important to mention that out of three extremely wet years, two (1992, 1997) were strong to very strong El Nino years, while two (2008, 2012) out of the three dry years were weak to moderate La Nina years [53]. In 1992, 52 mm of rainfall fell in Gizan and 40 mm in Jeddah on 2 and 11 November, respectively. In 1997, the total annual rainfall came from only a few days, 78 (37), 120 (41), and 105 (99) mm on 24, 25, and 29 March, respectively, in Abha (Khamis Mushait). This indicates that, in Saudi Arabia, extreme events contribute strongly to the total rainfall, as discussed later.

The monthly-average rainfall over Saudi Arabia is in the range of approximately 1 to 15 mm month$^{-1}$, with the highest variability during the wet season (October–May). The country-average annual cycle (black solid curve in Figure 5a) shows that the highest rainfall is observed during the wet season, while the lowest is during the dry season. The selected years with months (vertical bars) experienced the highest and lowest rainfall amounts (Figure 5a–f). From the analysis, it is evident that

the northern, coastal, and interior regions receive rainfall only during the wet season, and that extreme rainfall events occur during the early three months of the wet season (Figure 5b–d). The highland and southern regions receive the highest amount of rainfall, and experience the most extreme rainfall events, during March, April, and May (Figure 5e,f). However, the highland and southern regions are also vulnerable to extreme rainfall events during the dry season, particularly during August.

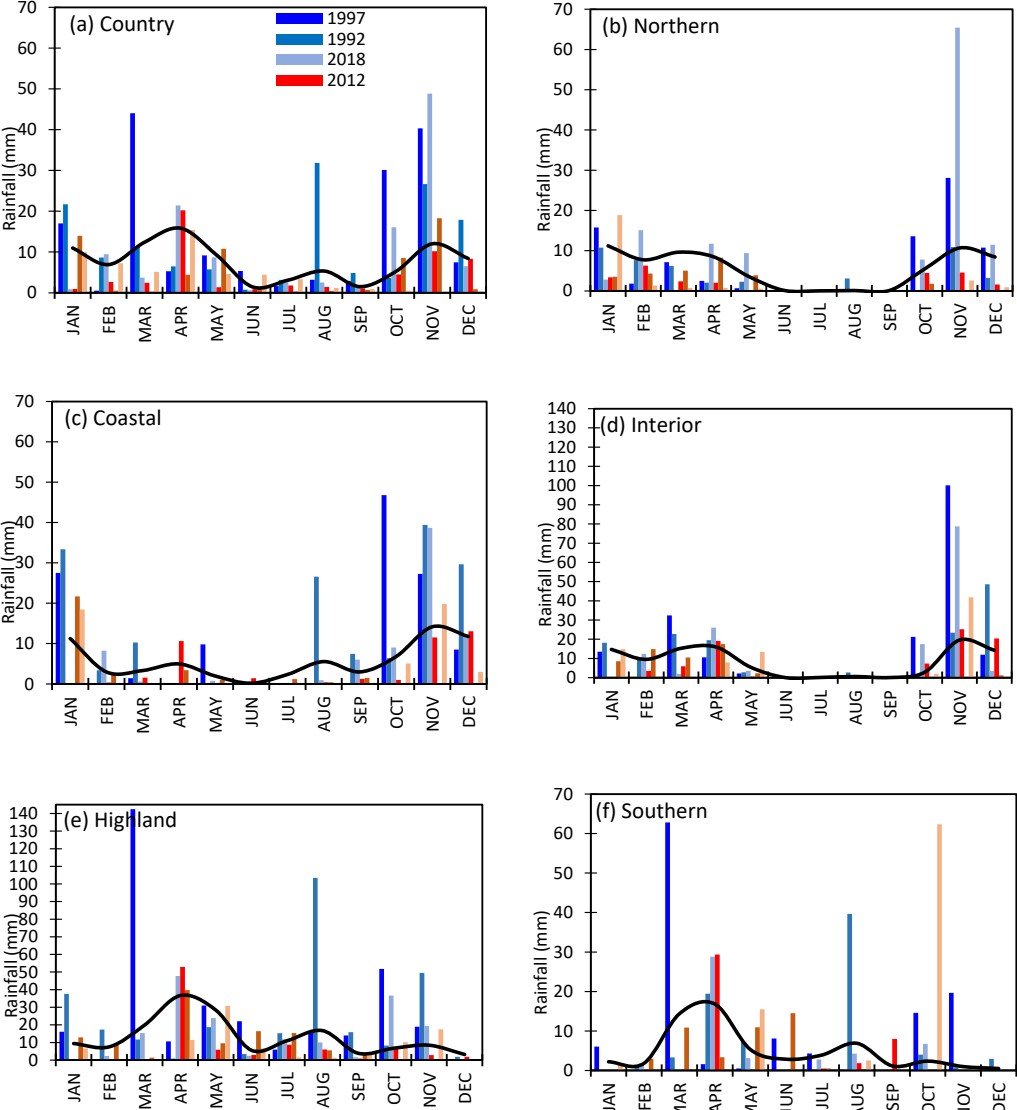

**Figure 5.** Annual cycle of rainfall (mm month$^{-1}$) for the (**a**) country, (**b**) northern, (**c**) coastal, (**d**) interior, (**e**) highland, and (**f**) southern regions for wet (1992, 1997, and 2018) and dry (2007, 2008, and 2012) years along with the rainfall climatology (mm, black solid curve) averaged for the period 1978–2019. The vertical axis scales are not the same for all panels.

Various characteristics were observed in rainfall changes at 25 stations across Saudi Arabia with respect to the base period. It was found that the rainfall decreased annually at most of the stations, while a rainfall increase was observed at some stations. For instance, the rainfall-change analysis shows that Hail (Turaif) experienced the significant lowest (highest) value of −164.9 (58.6) mm decade$^{-1}$ of annual rainfall changes (Figure 6a). However, Yenbo and Wejh observed almost no changes. Similar behavior of rainfall changes over the 25 stations was observed during the wet season (Figure 6b). In this season, a negligible change was observed over Jeddah, Najran, Yenbo, Al-Jouf, and Dammam. Interestingly, the wet season and annual statistics show positive changes over the northern parts, along with negative

changes over the central and southwestern parts of Saudi Arabia. This behavior shows the dominant role of Mediterranean troughs during the wet season. During the dry season, negligible changes were observed over most of Saudi Arabia, except at some stations in the southwestern mountainous region (Figure 6c). In this season, Abha was the only station that experienced an increase in rainfall of around 40 mm decade$^{-1}$. Overall, most stations recorded a rainfall decrease, while a few recorded an increase.

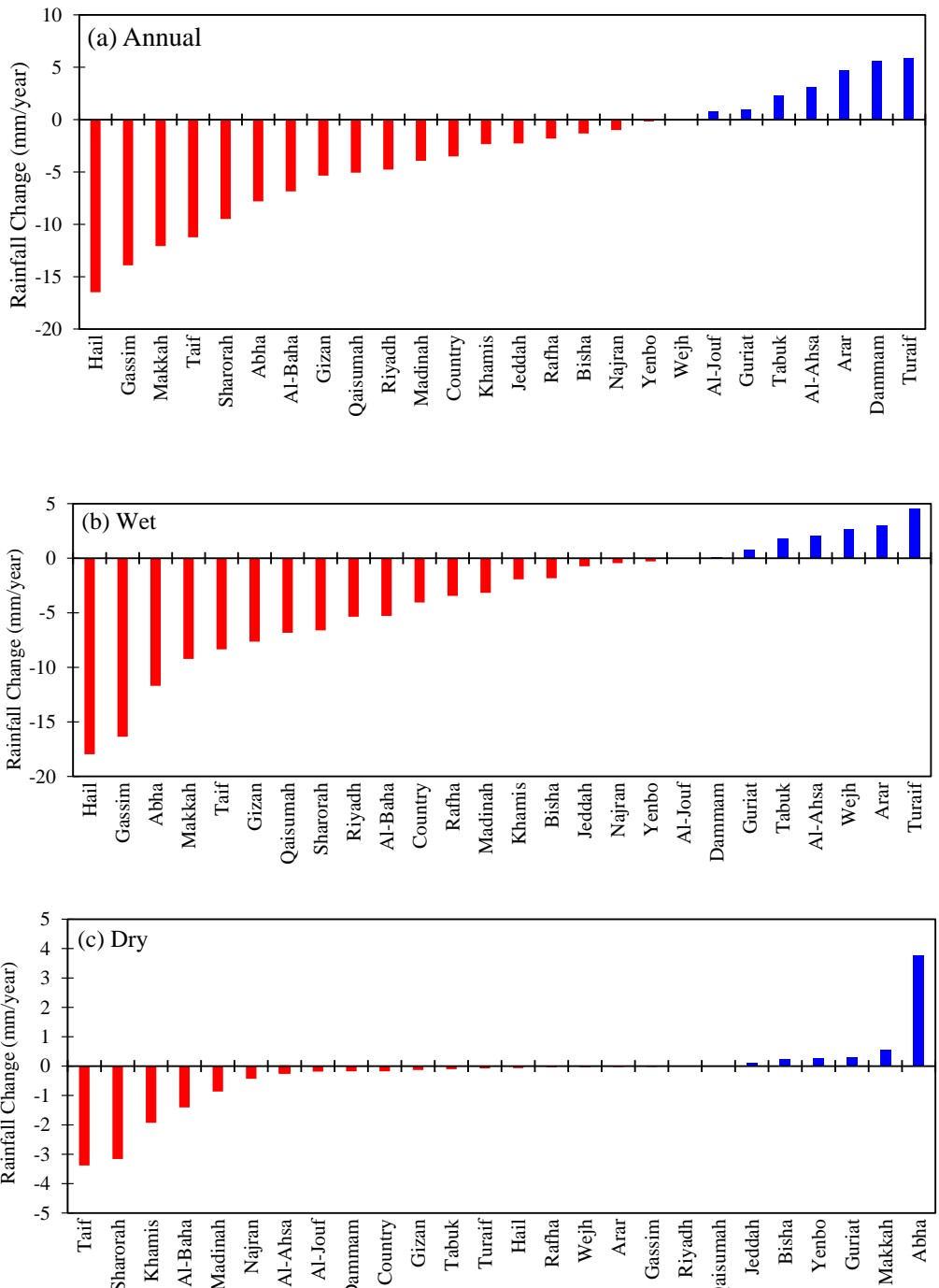

**Figure 6.** Changes in rainfall at 25 stations and over the entire country for (**a**) annual, (**b**) wet season (October–May), and (**c**) dry season (June–September) with respect to base period 1981–2010 averaged for the period 1978–2019. The blue and red bars indicate positive and negative changes, respectively. The scales on the vertical axes are not the same for all panels. The station locations in different climate zone can be seen in Table 1.

### 3.3. Extreme Rainfall Events Frequency and Trends

The mean climatology of the frequency of extreme rainfall events based on different thresholds ($\geq$1, $\geq$5, $\geq$10, $\geq$20, $\geq$30, and $\geq$50 mm) is shown in Figure 7. The spatial maps of rainfall frequency, calculated for each station, suggest that high-intensity events are observed mainly in the central and southwestern regions. However, some exceptions are reported in the northwestern and eastern regions, which usually have a low frequency of intense rainfall events. The high mountainous southwestern region experiences local convective activity due to orographic effects, which consequently cause rainfall extremes [54]. The spatial patterns of the frequency of daily rainfall for different thresholds show very similar behavior. The wet season rainfall is dynamical. A line of convergence often develops along with the central and southwestern parts of Saudi Arabia, which can trigger significant rainfall events. Therefore, during the wet season a lot of rain falls on the northeastern, central, and southwestern parts of Saudi Arabia. The spatial climatological maps of the frequency of extreme rainfall events show that intense rainfall events are most likely to occur in the southwestern region along the Red Sea coast (Figure 7e,f).

In addition to the annual frequency of extreme rainfall events with different thresholds, the trends of these frequencies are shown in Figure 8 for the period 1978–2019. Figure 8 shows significantly decreasing trends in the frequency of rainfall extremes in the central and southwestern parts of Saudi Arabia, which have the highest amount of annual rainfall. However, the northwestern, coastal, eastern, and southern stations all show increasing trends in the frequency of extreme rainfall events, particularly for intense events ($\geq$30, $\geq$40, and $\geq$50 mm). Interestingly, more stations have increasing trends for intense rainfall events than for low rainfall events.

Figure 9 shows the time series of extreme rainfall event frequency for various thresholds in Saudi Arabia during the period 1978 to 2019. It is clear from the figure that the high and low peaks for the different thresholds line up almost vertically. The maximum extreme rainfall event frequency for all the thresholds happened in 1982 (followed by 1997 and 2018), while the frequency dropped to its lowest level in 2007. Note that 1992, 1997, and 2018 are all El Nino years. Additionally, note how the extreme rainfall event frequency drops as the threshold is increased. The different threshold frequency ranges are the following: $\geq$1 (9–20 day year$^{-1}$), $\geq$5 (4–9 day year$^{-1}$), $\geq$10 (2–5 day year$^{-1}$), and $\geq$20 mm (1–3 day year$^{-1}$). For the thresholds $\geq$25 to $\geq$50 mm, the frequency remained under 2 day year$^{-1}$. Rainfall events with daily rainfall thresholds $\geq$1 and $\geq$5 mm are not necessarily extreme events. Rainfall events with threshold values of $\geq$25 mm and above have the lowest occurrence rate during the period 1978–2019. Rainfall events with a threshold of $\geq$25 mm may be considered as genuine extremes [2], and are investigated further below. The frequency time series results are consistent with the spatial distribution, as shown in Figure 7a–f. Interestingly, Figure 9 shows that the frequency of extreme events at all thresholds decreased significantly after 2000, though it increased again in recent years after 2010. This analysis shows that the greatest increase is likely to occur in short-duration rainfall. However, the analysis is potentially suggesting an increase in the magnitude and frequency of flash floods in future years because of the increasing trend of intense rainfall events.

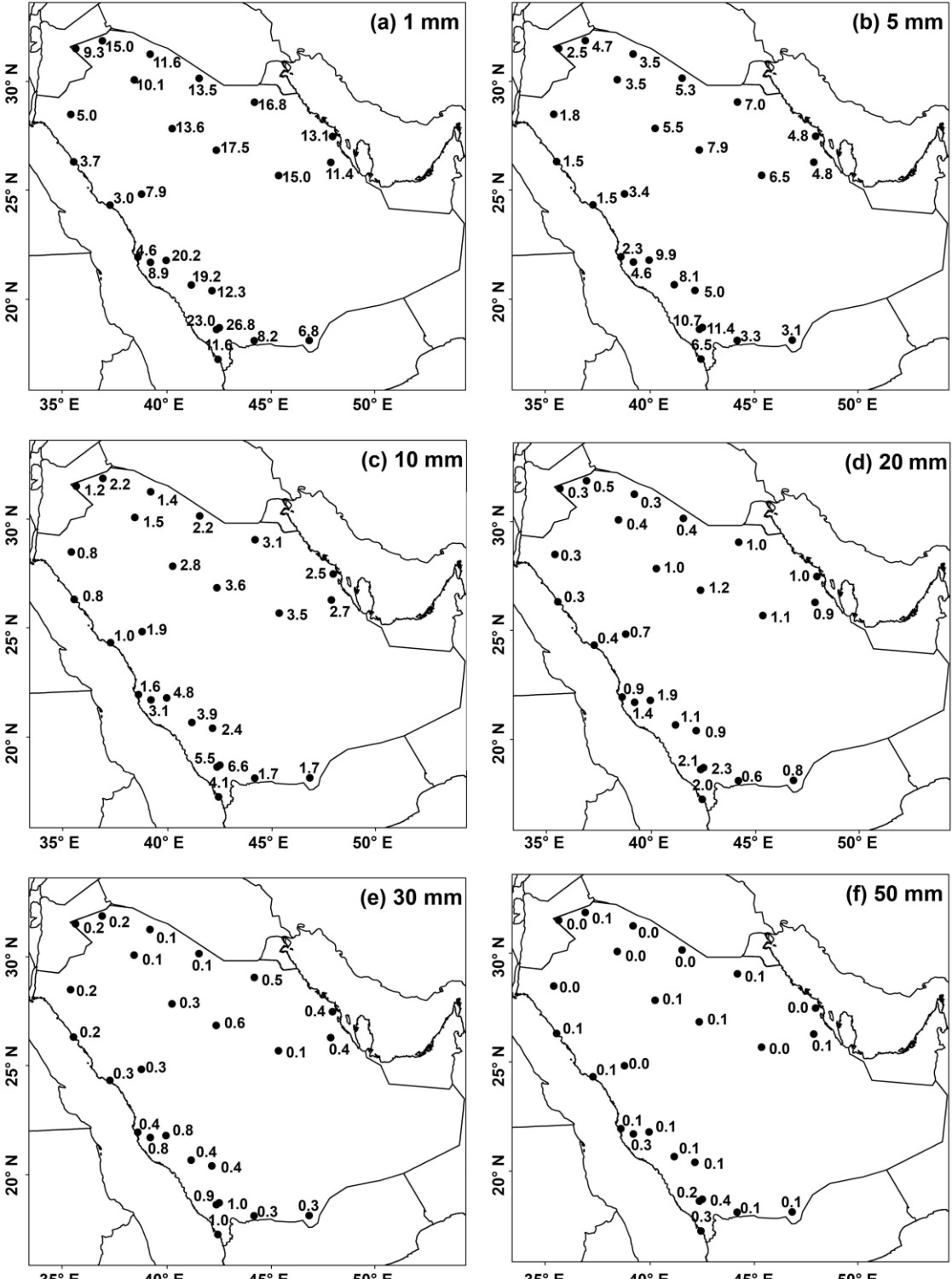

**Figure 7.** Annual frequency of rainfall extremes for thresholds: (**a**) 1, (**b**) 5, (**c**) 10, (**d**) 20, (**e**) 30, and (**f**) 50 mm for the period 1978–2019.

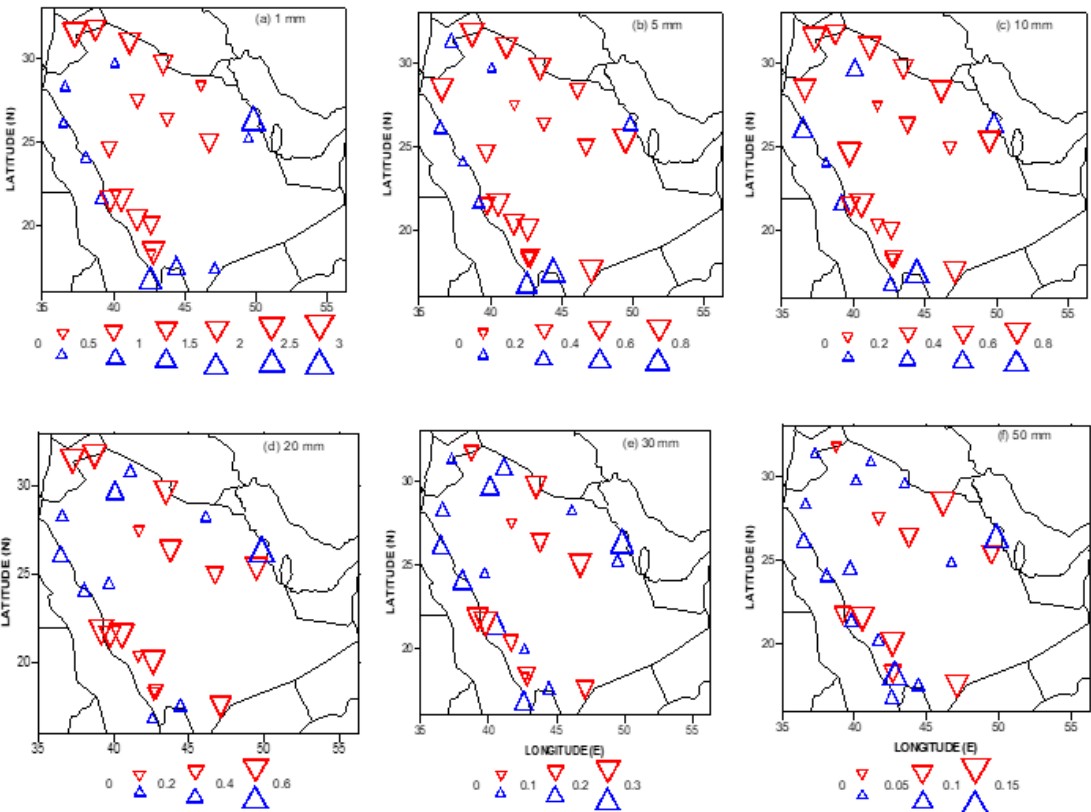

**Figure 8.** Trends (mm decade$^{-1}$) of rainfall extremes for thresholds: (**a**) ≥1, (**b**) ≥5, (**c**) ≥10, (**d**) ≥20, (**e**) ≥30, and (**f**) ≥50 mm for the period 1978–2019. The upward (blue) and downward (red) triangles represent increasing and decreasing trends, respectively. Note that the scales of all the panels are not the same.

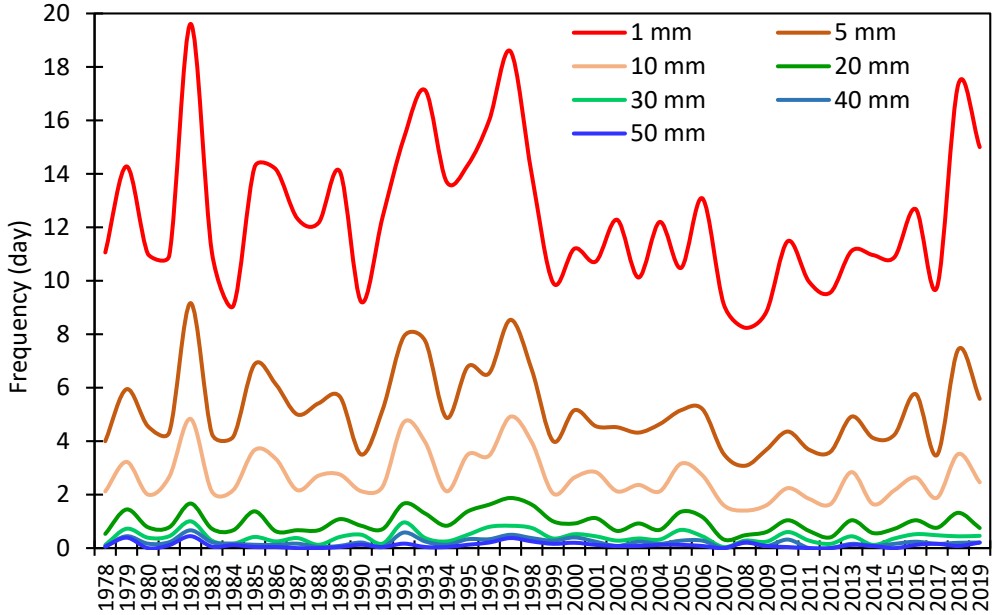

**Figure 9.** Extreme rainfall time sequence of event frequency for different thresholds in Saudi Arabia for the period 1978–2019.

The observational data indicate that most stations show negative trends in the frequency of extreme events that have rainfall thresholds of up to 50 mm day$^{-1}$ (Figure 10). For example, Dammam, Najran, and Gizan have significant positive trends in the frequency of less intense rainfall events, but the majority of stations showed significantly decreasing trends as the intensity increased. For the Abha, Al-Quasumah, and Gassim stations, the trends were negative for less intense rainfall but became positive for intense events. It is evident that rainfall events with light to moderate intensity show significantly decreasing trends over most of Saudi Arabia, but the majority of stations display positive trends for the most intense rainfall events. Overall, for the country average, trends in rainfall event frequency change from negative to positive as the event intensity increases. As the threshold increases, a large number of stations, which show a negative trend at a low threshold, move towards more positive trends, which is a clear indication that the country may experience more extreme rainfall events in the future.

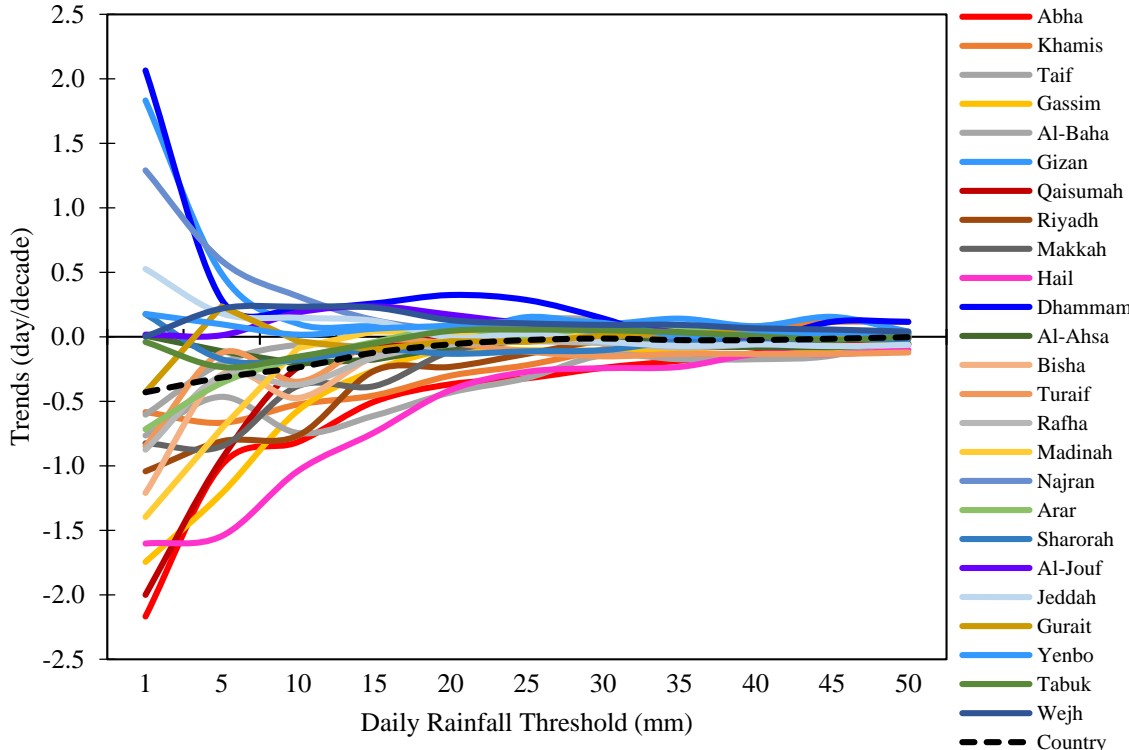

**Figure 10.** Extreme rainfall trends for different thresholds at each station in Saudi Arabia for the period 1978–2019.

Statistical analysis of the annual frequency of rainfall events reveals the decadal behavior of extreme rainfall events in five climatic regions of Saudi Arabia (Figure 11). There is a marginal increase in the frequency of extreme rainfall events in the most recent decade (Figure 11a). The frequency of extreme events is highly variable from north to south and has changed over time during the period 1978–2019. The stations in the northern region show an increase in the frequency of extreme rainfall events during the most recent decade (Figure 11b). The stations in the coastal and highland regions show the highest frequency of extreme rainfall events as compared to other regions (Figure 11c,e), and this may be due to the moisture convergence zones that occur there. The highest frequency of light to moderate rainfall events is observed in the highland region (Figure 11e). The stations in the southern region show relatively few very extreme rainfalls events (Figure 11f). The historical register shows that the frequency of extreme rainfall events is high in the highland, interior, and coastal regions as compared to the other two regions. From the analysis, it is concluded that the highland, interior, and coastal regions will experience more frequent extreme rainfall events in the near future.

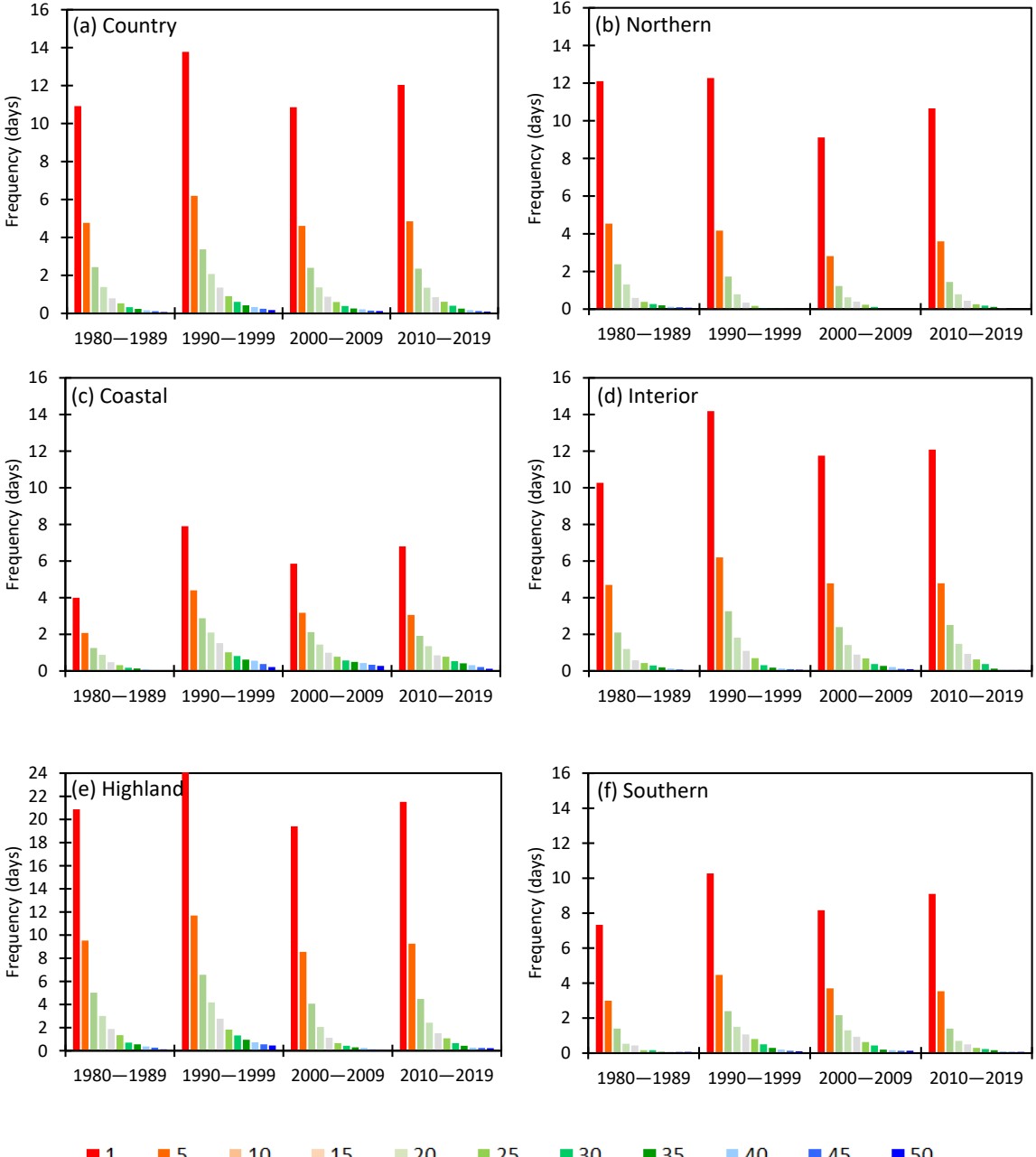

**Figure 11.** Annual frequency (days) of rainfall extremes in different decades for (**a**) country average, (**b**) northern, (**c**) coastal, (**d**) interior, (**e**) highland, and (**f**) southern regions in Saudi Arabia. The vertical scale for all panels is the same except for the highland.

The analysis of the time series of daily rainfall at 25 stations in Saudi Arabia showed a significantly decreasing trend in the frequency of light to moderate intensity rainfall events (Table 2). The trends in the extremes are observed to be homogeneous in the central and southwestern parts, with some differences in the significance levels. However, the daily rainfall data analysis shows increasing trends, though insignificant, in the frequency of extreme rainfall events of ≥30 mm. Any positive or increasing trend in the frequency of extreme rainfall events is a serious concern for the regions of Saudi Arabia vulnerable to flash floods. These statistical analyses can have severe implications for the probability of flash flooding events and droughts, which can be triggered under the influence of a future warmer climate [24].

**Table 2.** Rainfall trends (mm decade$^{-1}$) at different station locations in Saudi Arabia. The superscripts a, b, and c represents trend significance at the 90%, 95%, and 99% confidence level, respectively.

| Region | Station | 1 mm | 5 mm | 10 mm | 15 mm | 20 mm | 25 mm | 30 mm | 35 mm | 40 mm | 45 mm | 50 mm |
|---|---|---|---|---|---|---|---|---|---|---|---|---|
| Northern | Tabuk | −0.04 | −0.23 | −0.15 [c] | −0.05 | 0.05 | 0.06 | 0.05 | 0.03 | 0.00 | −0.02 | 0.00 |
| | Turaif | −0.85 | −0.12 | −0.35 | −0.05 | −0.05 | −0.12 | −0.15 [b] | −0.13 [b] | −0.13 [b] | −0.13 [b] | −0.12 [b] |
| | Gurait | −0.43 [a] | 0.24 [a] | −0.03 | −0.10 | −0.03 | −0.04 | 0.01 | 0.03 | 0.04 | 0.00 [a] | 0.00 [a] |
| | Rafha | −0.88 | −0.30 | −0.37 [c] | −0.16 | −0.06 | −0.04 | −0.05 | −0.08 [b] | −0.06 [c] | 0.00 | 0.00 |
| | Arar | −0.72 | −0.36 | −0.17 | −0.04 | 0.07 | 0.00 | 0.08 [b] | 0.07 [b] | 0.03 | 0.00 | 0.00 |
| | Al-Jouf | 0.02 | 0.01 | 0.19 | 0.24 [c] | 0.17 [b] | 0.11 [b] | 0.10 [a] | 0.03 [c] | 0.00 | 0.00 | 0.00 |
| | Hail | −1.60 [b] | −1.54 [a] | −1.04 [a] | −0.74 [a] | −0.41 [a] | −0.27 [a] | −0.24 [a] | −0.23 [a] | −0.13 [a] | −0.13 [a] | −0.10 [b] |
| Interior | Gassim | −1.74 [b] | −1.21 [b] | −0.57 [c] | −0.26 | −0.10 | −0.10 | −0.11 | −0.11 | −0.15 [b] | −0.10 [c] | −0.05 |
| | Riyadh | −1.04 | −0.81 | −0.76 [c] | −0.26 | −0.23 | −0.13 | −0.03 | −0.02 | −0.02 | −0.02 | 0.00 |
| | Al-Qaisumah | −2.00 [a] | −0.95 [b] | −0.23 | −0.10 | 0.02 | 0.03 | 0.00 | 0.02 | 0.03 | −0.03 | −0.01 |
| | Dammam | 2.06 | 0.29 | 0.22 | 0.26 | 0.32 | 0.29 | 0.14 | −0.04 | −0.04 | 0.12 | 0.12 |
| | Al-Ahsa | 0.01 | −0.11 | −0.21 | −0.18 | −0.08 | 0.00 | 0.03 | −0.08 | −0.08 | −0.08 | −0.05 |
| | Madinah | −1.40 [b] | −0.71 [b] | −0.09 | 0.04 | 0.03 | 0.05 | 0.01 | 0.03 | 0.01 | 0.05 | 0.04 |
| Coastal | Wejh | 0.00 | 0.22 | 0.23 [c] | 0.23 [b] | 0.13 | 0.10 | 0.09 | 0.09 | 0.07 | 0.06 | 0.04 |
| | Yenbo | 0.18 | 0.10 | 0.02 | 0.06 | 0.09 | 0.12 [c] | 0.11 [c] | 0.01 | 0.02 | 0.03 | 0.03 |
| | Jeddah | 0.53 | 0.18 | 0.15 | 0.12 | −0.01 | 0.02 | −0.04 | −0.07 | −0.06 | −0.07 | −0.06 |
| | Makkah | −0.83 | −0.85 [b] | −0.38 | −0.38 [c] | −0.11 | −0.03 | −0.02 | 0.02 | 0.00 | −0.02 | 0.03 |
| | Gizan | 1.83 [a] | 0.49 | 0.10 | 0.08 | 0.01 | 0.15 | 0.10 | 0.14 | 0.08 | 0.15 | 0.04 |
| Highland | Taif | −0.60 | −0.20 | −0.06 | −0.08 | −0.03 | 0.13 | 0.12 | −0.01 | −0.03 | −0.04 | −0.01 |
| | Al-Baha | −0.76 | −0.46 | −0.75 [c] | −0.61 [b] | −0.43 [b] | −0.32 [a] | −0.14 | −0.17 [b] | −0.17 [b] | −0.15 [c] | 0.02 |
| | Khamis Mushait | −0.58 | −0.67 | −0.52 | −0.45 [c] | −0.30 | −0.22 | −0.10 | 0.03 | 0.08 | 0.12 | 0.12 [c] |
| | Abha | −2.17 [a] | −0.99 [b] | −0.82 [b] | −0.50 | −0.37 | −0.32 | −0.24 | −0.18 | −0.13 | −0.08 | −0.06 |
| Southern | Bisha | −1.21 [c] | −0.29 | −0.47 [c] | −0.15 | −0.04 | −0.03 | 0.01 | 0.04 | −0.01 | −0.01 | −0.01 |
| | Najran | 1.29 [c] | 0.59 [c] | 0.31 | 0.13 | 0.06 | 0.07 | 0.04 | 0.00 | 0.04 | 0.02 | 0.02 |
| | Sharorah | 0.18 | −0.17 | −0.18 | −0.10 | −0.13 | −0.11 | −0.10 | −0.03 | −0.03 | −0.02 | −0.02 |
| | Country | −0.43 | −0.32 [c] | −0.24 [c] | −0.12 | −0.06 | −0.02 | −0.01 | −0.02 | −0.03 | −0.01 | 0.00 |

For the lower event thresholds (1–20 mm), the rainfall trend is decreasing over many stations, while a few stations show an increasing trend (Figure 12). For the higher thresholds (25–50 mm), this scenario is reversed, and a large number of stations show an increasing trend of extreme rainfall, while a few stations have decreasing trends, except for the ≥45 mm threshold. Overall, there is an increase in the trend of extreme rainfall events with an intensity of ≥25 mm and above. For the threshold ≥50 mm, 15 stations (60%) show a positive trend, while 14 stations (56%) show a positive trend for the threshold ≥30 mm. The northern and coastal regions contributed 33% and 27%, respectively, of the 60% positive trend for the threshold ≥50 mm. For light rainfall, more stations show a decreasing trend than an increasing trend. In the case of the threshold ≥1 mm, 64% of the stations show a decreasing trend, of which the northern region contributes 38%, while the interior and highland regions contribute 25%. These are all evidence of an increasing trend in extreme rainfall events in Saudi Arabia. The extreme rainfall trends and events are essential for the definition of the regional climate. This kind of analysis will be helpful to understand the risk for the country and to take possible countermeasures.

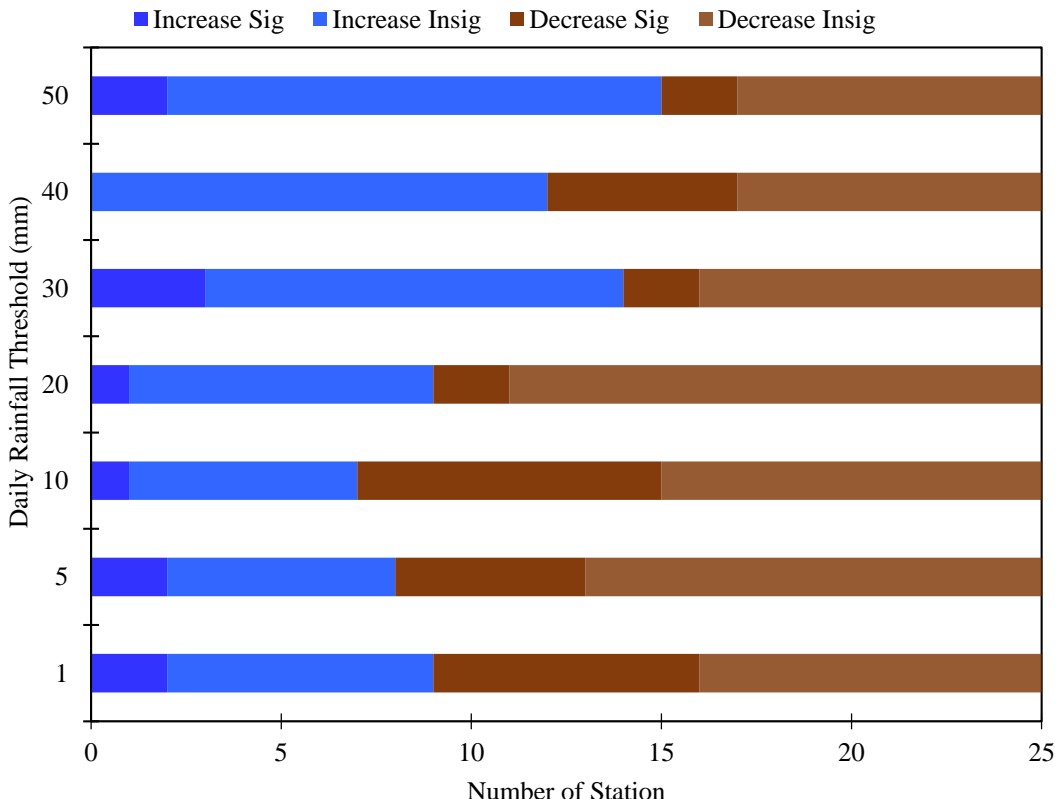

**Figure 12.** The number of stations with extreme rainfall trends increasing (blue) and decreasing (brown) at different thresholds for the period 1978–2019.

### *3.4. Rainfall Return Period*

The depth–duration–frequency (DDF) curves are plotted on a logarithmic scale for the vertical axis only (Figure 13). Abha and Taif receive almost 100 mm of rainfall every year, and more than 550 mm is expected to occur within the next 40-year return period (Figure 13a). However, Sharorah, Jeddah, and Yenbo showed that every year rainfall is observed below 1 mm, and within the mentioned 40-year return period only 50 mm of rainfall is expected to happen. For the wet season return period curves (Figure 13b), the Abha, Gassim, and Al-Qaisumah stations observed around 20 mm of rainfall every year, and within the mentioned 40-year return period, around 250 mm is expected to happen. However, Sharorah, Jeddah, Yenbo, and Tabuk observed rainfall of below 1 mm each year, and within the mentioned 40-year return period 50 mm of rainfall is expected to happen. Finally, for the dry season return period (Figure 13c), Abha, Al-baha, Khamis Mushait, and Taif typically observed rainfall of between 1 to 10 mm, and a 100 mm rainfall event is expected to happen at some time in the next 40 years. There are many stations where rainfall does not occur every year in the dry season. In such stations, some rainfall may occur within a 5-year period. For example, at Al-Jouf, Turaif, and Yenbo, a 1 mm rainfall event is expected to occur sometime within a 5-year return period.

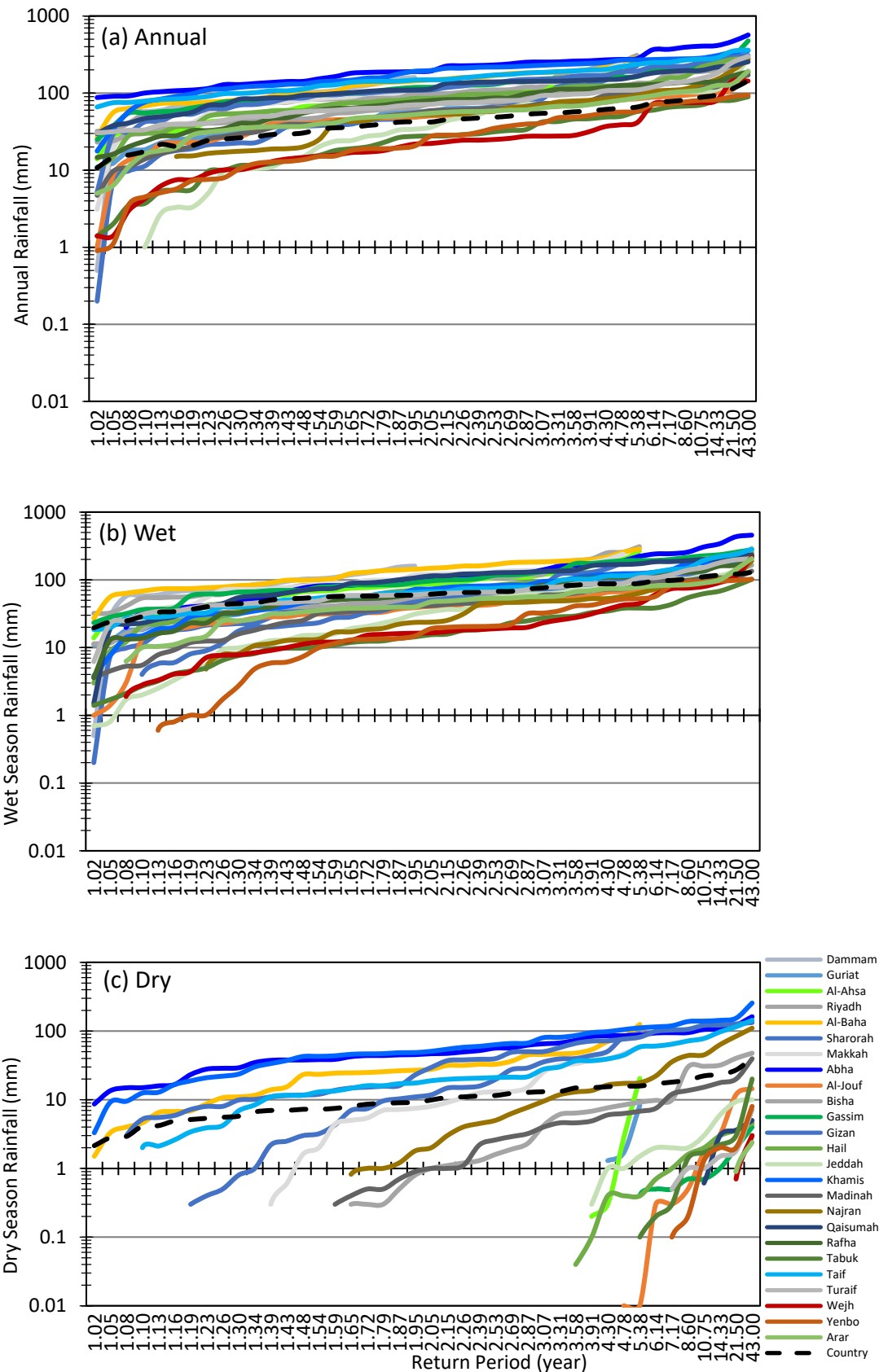

**Figure 13.** The depth–duration–frequency curves of the return periods for the (**a**) annual, (**b**) wet season, and (**c**) dry season rainfall at different stations averaged for the period 1978–2019. Rainfall amounts on the vertical axis are in a logarithmic scale.

### 3.5. Contribution of Extreme Rainfall to the Total Rainfall

The contribution of extreme rainfall to the total rainfall is important for the development of early warning systems and for planning preventive measures against flash floods within the country. For the 90th, 95th, and 99th percentiles, it was found that the amount of rainfall at the 90th percentile was greater, while it was less for the 99th percentile as compared to the threshold-based rainfall (Figure 14a). For simplicity, only three threshold-based rainfall amounts are shown here. The threshold-based rainfall amounts and percentile-based amounts are different for different stations. However, the rainfall amount at the 95th percentile is almost equivalent to rainfall with a threshold of ≥26 mm at the country scale. The scatter plot (Figure 14b) clearly shows that the country-averaged rainfall amounts for the 95th percentile and the ≥26 mm threshold lies along the 45-degree line of the plot. For the 95th percentile, the threshold of ≥20 mm allows too much rain, while the threshold of ≥30 mm permits too little. Therefore, the threshold of ≥26 mm was chosen as the extreme rainfall threshold (close to the 95th percentile) over the country as a whole.

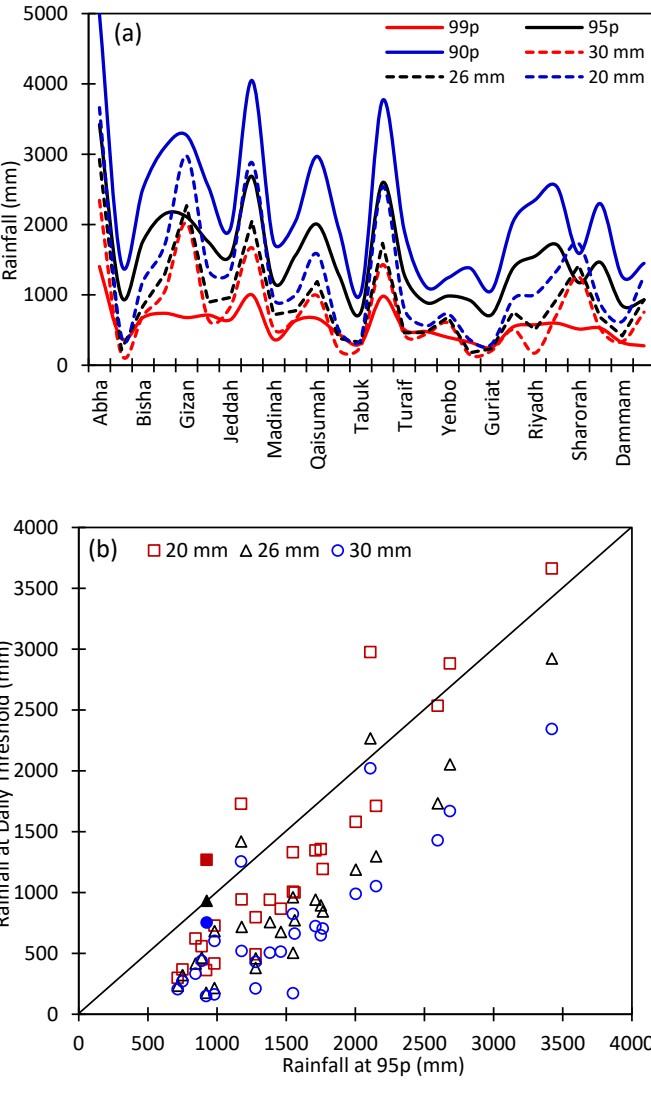

**Figure 14.** (**a**) Amount of rainfall (mm) at the 90th, 95th, and 99th percentiles along with at the ≥20, ≥26, and ≥30 mm thresholds in Saudi Arabia for the period 1978–2019. (**b**) Scatter plot of the percentile-based and threshold-based rainfall. The open-square, triangle, and circle represent station values, while the closed square, triangle, and circle represent the country average at ≥20, ≥26, and ≥30 mm, respectively.

Given the fact that extreme rainfall events contributed to the total rainfall in Saudi Arabia, a comprehensive analysis was performed on the daily rainfall data to explore the changes in the total amount of water as extreme rainfall as well as quantifying the contributions from changes in the intensity and the frequency (Figure 15). The investigation of extreme rainfall events shows that extreme rainfall events of ≥26 mm contributed between 8% (Arar) and 50% (Yenbo) of the total rainfall occurring in Saudi Arabia. Annually, less than 10% of extreme events contribute 26% of the total rainfall (24 mm), while more than 90% of the events contribute the other 74% (66 mm) for the entire country (Figure 15a). For the country average, rainfall occurred mostly in the wet season, and extreme events contributed the most (32% of the total) in November, followed by 30% in March.

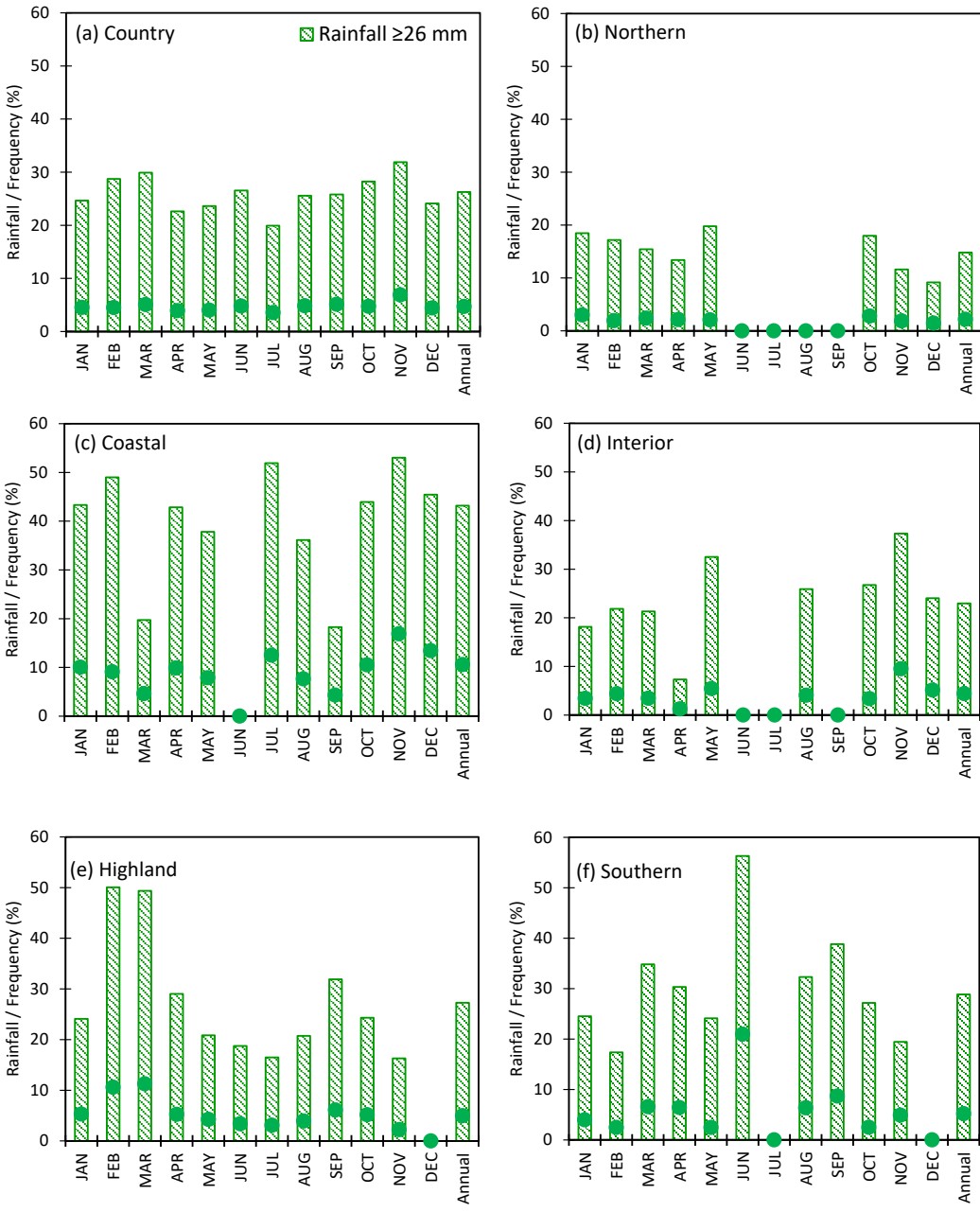

**Figure 15.** Contribution of extreme rainfall to total amounts (%), and extreme event frequency (%) at the ≥26 mm threshold in Saudi Arabia and five climate regions for the period 1978–2019. Annually, fewer than 10% of extreme events contributed 26% of the total rainfall (24 mm), while more than 90% of events contributed the other 74% (66 mm) for the entire country. (**a**) Country. (**b**) Northern. (**c**) Coastal. (**d**) Interior. (**e**) Highland. (**f**) Southern.

For the northern region, rainfall is almost zero during the dry season, and extreme events contribute the most (20%) in May, followed by 19% in January (Figure 15b). In the coastal region, extreme events contribute strongly, with the largest contributions in the October–February period (Figure 15c). In this region, extreme events contribute, on average, 47% (43–53%) of the total in each month from October to February, with the highest (53%) in November. The inland region follows a similar pattern to the northern region (Figure 15d). For the highlands, rainfall occurs in almost all months, especially in March–May and July–August (Figure 15e). In this region, extreme rainfall events contribute the most (50% of the total) in February, followed by 49% in March. The rainfall characteristics in the southern region are different from in other regions, since rainfall occurs mainly in the March–May period (Figure 15f). In this region, extreme rainfall has the highest contribution (56%) in June. Hence, the contribution of extreme events to the total rainfall amount fluctuates from region to region, with the highest contribution (up to 56%) found in the southern region in June, while the largest contribution from the coastal region comes between October and February. Among the five climate regions, the coastal region is where extreme events comprise the largest fraction of the total rainfall.

## 4. Summary and Conclusions

An analysis of the observational data for the 42-year period 1978–2019 has been carried out to understand the patterns of rainfall climatology and the frequency of extreme rainfall events over Saudi Arabia. The contribution of extreme rainfall events to the total amount of rainfall was also examined by analyzing the daily rainfall records in five climate regions of Saudi Arabia. The observational dataset shows that the highest amounts of rainfall occur during the wet season (October–May) and the lowest in the dry season (June–September). Climatologically, the greatest rainfall intensity occurs in the central and southwestern parts along the Red Sea coast. The northern, northwestern, and southeastern areas have a mean climatology that ranges from low to medium intensity.

In general, the annual and wet seasonal rainfalls over Saudi Arabia reveal a decreasing trend during the study period. The highest rate of decrease in the annual (and wet-season) rainfall has accelerated to between 10 to 20 mm decade$^{-1}$ in the central and southwestern region, which is an alarming situation for the water resources, agricultural production, and socio-economic needs of the country. However, some stations in the eastern, northwestern, and southern parts show weak (not statistically significant) increasing trends. There are significant negative rainfall trends at annual and seasonal time scales over several stations for the entire study period (1978–2019). However, the trends in the dry season are small and insignificant.

The contribution of extreme events to the total rainfall amount is important for application-oriented tasks. This analysis found that the ≥26 mm threshold is the appropriate country-averaged cut-off for defining extreme events. The contribution of extreme events to the total rainfall varies from month to month as well as from one region to another. The highest contribution (up to 56%) was found in the southern region in June, where the maximum contribution comes from coastal regions. Regionally, coastal regions contributed about 47% in each month from October to February, which is important for effective planning. For the country, extreme rainfall contributes the most (52% of the total) in November and the least (20% of the total) in July. Overall, extreme rainfall contributes between 8% (Arar) and 50% (Yenbo) of the total rainfall in Saudi Arabia.

Further, it is found that there is an increasing trend in the frequency of extreme rainfall events, though this is statistically insignificant for many stations. This increasing trend in the frequency of extreme rainfall over Saudi Arabia may be associated with climate change over recent decades, which needs further investigation to understand the impact of climate change over the country. The analyses in this study have determined trends in extreme rainfall events which will be helpful for deciding appropriate mitigation measures, in particular for the water and agriculture sectors of the country. The identification of homogeneous and heterogeneous trends provides a basis for making predictions of these events. Future changes in the extreme rainfall trends in Saudi Arabia can be further explored using climate model simulations.

**Funding:** This research received no external funding.

**Acknowledgments:** The author thank the three anonymous reviewers for their valuable comments and suggestions. The author also acknowledges the Centre of Excellence for Climate Change Research (CECCR), King Abdulaziz University, for supporting this research work. The GAMEP is acknowledged for the observational dataset, and CHIRPS and GPCC are acknowledged through their website. The computations described in this paper were performed on the Aziz Supercomputer at King Abdulaziz University's High-Performance Computing Center, Jeddah, Saudi Arabia.

**Conflicts of Interest:** The authors declare no conflict of interest.

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
