# Peer review of "Rainfall Trends and Extremes in Saudi Arabia in Recent Decades"

_atmosphere, doi:10.3390/atmos11090964_

Round 1

Reviewer 1 Report

Paper:  Rainfall trends and extremes in Saudi Arabia in the 2 recent decades

The paper analyses rainfall records in Saudi Arabia over the period 1978-2019, in terms of trends and extreme events.

The study does not present any novelty, and the methodologies are neither new nor adequately explained:

  • “The rainfall trends and their significance levels were calculated using the regression equation and F-test, respectively”: what is the confidence level considered? The precision is given only in the tables legend. Besides, non parametric tests are prefered in climate studies because the data are neither independent nor normally distributed.
  • Most of the trends are not significant over the entire period, thus 10-year trends have very low chance to be significant.
  • The base period encompasses most of the observation period thus changes according to the base period are rather anomalies; the mean anomaly over the period cannot be used to infer a trend.
  • How exactly are the Return Levels estimated? It seems that when trends are identified, some extrapolation is made to the future, how? Why the statistical extreme value theory is not used? For sure 46 years may be too short to use the block maxima approach, but for the rainiest seasons, Peak Over Threshold could be tested.
  • The conclusion “This increasing trend in the frequency of extreme rainfall over Saudi Arabia may be associated with the climate change over recent decades.” cannot be drawn from the analysis. As mentioned, interannual variability is large, the observation period is quite short, it is thus impossible to attribute the identified changes (which, by the way, are not always significant) to climate change without any further investigation using more data and climate model simulations.

Author Response

Reviewer 1

Comments and Suggestions for Authors

Paper: Rainfall trends and extremes in Saudi Arabia in recent decades

The paper analyses rainfall records in Saudi Arabia over the period 1978-2019, in terms of trends and extreme events.

Response: Thank you very much for your valuable comments and suggestion. This helped me to improve the quality of the manuscript.

The study does not present any novelty, and the methodologies are neither new nor adequately explained:

  • “The rainfall trends and their significance levels were calculated using the regression equation and F-test, respectively”: what is the confidence level considered? The precision is given only in the tables’ legend. Besides, non-parametric tests are preferred in climate studies because the data are neither independent nor normally distributed.

Response: Thanks to the reviewer for this comment. I agree with the reviewer’s comment and performed additional analysis using a non-parametric Mann-Kendall test for slope and significance of the trends. This is added in the revised version. The level of significance for regression analysis is also included in the revised version. I included both types of trends in the revised version so that it can be used for any comparative study in the future. Accordingly, the relevant text is also revised. Based on this new analysis, the following text has been added to the revised manuscript:

Methods (Section 2.2): The entire time series was divided into two halves periods, and the trend for each half was then computed along with the decadal trends. The frequency of rainy days was counted at each station for several threshold values, such as ≥1 mm, ≥5 mm up to ≥50 mm, with an interval of 5 mm. In this article, an extreme rainfall event is defined using the daily accumulated rainfall. An extreme rainfall event is defined as ‘An event having at least one day accumulated rainfall equal to or above a specific threshold’. The rainfall trends and their significance levels were calculated using the regression equation and F-test, respectively [40]. In addition to the above, a non-parametric trend analysis such as the Mann-Kendall (MK) trend test statistics [45,46] with MK statistic, S test statistic [47] has been used to obtain slope and significance of trends. There are also modified Mann-Kendall versions; however modified version has almost no change for rainfall and its extremes [17].

Results (Section 3.2): Because the rainfall data are neither independent nor normally distributed. Therefore the slope and significance of trends are also tested using Mann-Kendall test. According to this test, the slopes are showing increasing and decreasing trends similar to regression analysis with slight variation in magnitude. For example, the trends are -3.90, -8.13, 44.23, -32.40, and 33.57 mm decade-1 for the entire period, the first decade, second decade, third decade and last decade respective at annual scale. During the wet season, the trends are -5.96, -12.35, 54.17, -32.08, and 18.07 mm decade-1 for the entire period, the first decade, second decade, third decade and last decade respective. In dry season, trends are 6.61, 14.62, and -7.14 mm decade-1 for the first, second and third decade respectively while closer to total period and last decade, which are too small. Note that all of them show an insignificant level of confidence in Mann-Kendall test. As mentioned earlier that Mann-Kendall versions have no change in rainfall and its extremes, however very sensitive to temperature extremes [17]. Because this article is not focused on temperature extremes and Mann-Kendall test shows almost similar trend to regression analysis. Therefore trends of rainfall and extremes are evaluated using the regression method for the rest of the analysis.

  • Most of the trends are not significant over the entire period, thus 10-year trends have very low chance to be significant.

Response: Yes, this is true because the annual mean rainfall over Saudi Arabia is small and display large variations on different time scales. We agree with the reviewer that 10-year period is short; therefore, following the reviewer’s comment, we used 20-year period for trends in the revised version.

  • The base period encompasses most of the observation period thus changes according to the base period are rather anomalies; the mean anomaly over the period cannot be used to infer a trend.

Response: Thanks for this comment. Anomaly is calculated w.r.t base period 1981-2010, not w.r.t entire period 1978-2019.

  • How exactly are the Return Levels estimated? It seems that when trends are identified, some extrapolation is made to the future, how? Why the statistical extreme value theory is not used? For sure 46 years may be too short to use the block maxima approach, but for the rainiest seasons, Peak Over Threshold could be tested.

Response: Thanks to the reviewer for this comment. The 46-years period was not enough to use block maxima approach of the extreme value theory. Saudi Arabia receives very low annual mean precipitation. As we investigate trends in extremes during both wet and dry seasons, to what extent the EVT approach of Peak Over Threshold is valid during dry season is not clear. Therefore, instead of using EVT approach, we defined extremes by using percentile based threshold values. The procedure of Return Levels calculation is provided in the methodology section of the revised version as follows:

The return period of rainfall occurrence is defined as the time duration expectation over which there are no extreme event occurrences on the average. In practical works, generally, this duration is adapted as 2-year, 5-year, 10-year, 25-year, 50-year, or 100-year corresponding to 0.50, 0.20, 0.10, 0.04, 0.02, or 0.01 risk levels, respectively. As a method, the extreme event occurrences are assumed to take place independently from each other, and therefore, the probability distribution function (PDF) models provide a scientific basis. There are three distinct approaches as the (i) annual maxima, (ii) over a given threshold extremes, and (iii) threshold selection such that the number of years is equal to the number of extreme events over the threshold level [48]. The assessment of heavy rainfall events for certain return periods is often mandatory for the appropriate design of urban drainage and infrastructure, as well as for long-term planning. Robust assessments can only be obtained from the analysis of frequency of daily rainfall over long time-series. The daily total rainfall time-series is analyzed with the help of the gamma distribution. The anticipated annual, wet- and dry-season maximum rainfall during 42-year period are ascertained for different return periods at each station. The return periods are derived from the statistical distribution for every station from the annual extreme time-series. Where the trend is evident in the rainfall regime, a substitute for the concept of return period is suggested, based on the probability of occurrence of an extreme event of specified magnitude, during a period extending into the future during which the observed trend can be assumed to persist.

  • The conclusion “This increasing trend in the frequency of extreme rainfall over Saudi Arabia may be associated with the climate change over recent decades.” cannot be drawn from the analysis. As mentioned, interannual variability is large, the observation period is quite short, it is thus impossible to attribute the identified changes (which, by the way, are not always significant) to climate change without any further investigation using more data and climate model simulations.

Response: Agreed and rephrased the sentence as follows:

This increasing trend in the frequency of extreme rainfall over Saudi Arabia may be associated with climate change over recent decades, which needs further investigation to understand the impact of climate change over the country.

Submission Date: 24 July 2020

Date of this review: 30 Jul 2020 11:00:41

Reviewer 2 Report

Review report for manuscript id “atmosphere-893402”.

Overall, the paper is well written and the methodology is scientifically sound. However, I have a major concern about this paper. Self-citation in this article is very high. The author cited his work 24 times out of 55 references available. This self-citation is not acceptable.

The tables in this paper should be in landscape-oriented page.

Figure 2 and 8: the plots are overlapped. Please, replot and use black color font in figure 2 as the remaining figures instead of the gray color.

There are many extra spaces in several lines as in L62, L84, 113, etc. Please, check the whole paper and remove. On the contrary, many spaces between words are required as in L71 after “[25]”.

L52: replace the link with a reference.

L53: extra spaces after “which is a”. please, remove.

L56: How preserving national heritage sites and tourism are relevant to the analysis of rainfall extreme records?

L61: It is better to replace this line about south-east Asia with more related region or nearby countries to KSA. Many related studies can be reviewed here better than reviewing and citing a study which is not related and far away from your study area as

El Kenawy, A.M. and McCabe, M.F. (2016), A multi‐decadal assessment of the performance of gauge‐ and model‐based rainfall products over Saudi Arabia: climatology, anomalies and trends. Int. J. Climatol., 36: 656-674. doi:10.1002/joc.4374

Nashwan, M.S., Shahid, S. & Abd Rahim, N. Unidirectional trends in annual and seasonal climate and extremes in Egypt. Theor Appl Climatol 136, 457–473 (2019). https://doi.org/10.1007/s00704-018-2498-1

Nashwan, M.S., Shahid, S. Spatial distribution of unidirectional trends in climate and weather extremes in Nile river basin. Theor Appl Climatol 137, 1181–1199 (2019). https://doi.org/10.1007/s00704-018-2664-5

L73: It may be very clear to the author. But in the scientific community everything is debatable. Thus, remove “clear” from this line.

L110: Table 1 should be mention directly after this section. Now it is several pages away from where it was first mentioned. Same also with Figure 1.

L162: a comma is missing after “First”.

Author Response

Reviewer 2

Comments and Suggestions for Authors

Review report for manuscript id “atmosphere-893402”.

Overall, the paper is well written and the methodology is scientifically sound. However, I have a major concern about this paper. Self-citation in this article is very high. The author cited his work 24 times out of 55 references available. This self-citation is not acceptable.

Response: Thanks to the reviewer for the encouraging comments. The self-citation is reduced in the revised version though very few literature are available on the climate analysis in Saudi Arabia except the author and his research group work.

The tables in this paper should be in landscape-oriented page.

Response: Thanks for this comment. The Table format is according to the journal style as set up in the revised version.

Figure 2 and 8: the plots are overlapped. Please, replot and use black color font in figure 2 as the remaining figures instead of the gray color.

Response: Followed the suggestion. The overlaps are cleared. The font color of Fig. 2 is now in black instead of gray.

There are many extra spaces in several lines as in L62, L84, 113, etc. Please, check the whole paper and remove. On the contrary, many spaces between words are required as in L71 after “[25]”.

Response: Followed the suggestion and cleared the extra spaces.

L52: replace the link with a reference.

Response: This is from the website site and formatted according to the journal style.

L53: extra spaces after “which is a”. please, remove.

Response: This is done.

L56: How preserving national heritage sites and tourism are relevant to the analysis of rainfall extreme records?

Response: Extreme rainfall impacts on tourism and heritage sites.

L61: It is better to replace this line about south-east Asia with more related region or nearby countries to KSA. Many related studies can be reviewed here better than reviewing and citing a study which is not related and far away from your study area as

El Kenawy, A.M. and McCabe, M.F. (2016), A multi‐decadal assessment of the performance of gauge‐ and model‐based rainfall products over Saudi Arabia: climatology, anomalies and trends. Int. J. Climatol., 36: 656-674. doi:10.1002/joc.4374

Nashwan, M.S., Shahid, S. & Abd Rahim, N. Unidirectional trends in annual and seasonal climate and extremes in Egypt.  Theor Appl Climatol 136, 457–473 (2019). https://doi.org/10.1007/s00704-018-2498-1

Nashwan, M.S., Shahid, S. Spatial distribution of unidirectional trends in climate and weather extremes in Nile river basin. Theor Appl Climatol 137, 1181–1199 (2019). https://doi.org/10.1007/s00704-018-2664-5

Response: Followed the suggestion. The southeast Asia citation is removed, and the suggested citations are added.

L73: It may be very clear to the author. But in the scientific community everything is debatable. Thus, remove “clear” from this line.

Response: This is done.

L110: Table 1 should be mention directly after this section. Now it is several pages away from where it was first mentioned. Same also with Figure 1.

Response: Followed the suggestion. Table 1 and Figure 1 are moved in the suggested places.

L162: a comma is missing after “First”.

Response: Thanks, and this is done.

Submission Date: 24 July 2020

Date of this review: 05 Aug 2020 05:58:29

Reviewer 3 Report

Comments and Suggestions for Authors:

I read the article with interest. It is written clearly, in the proper order of information and is understandable to the reader. The analysis covers the period 1978-2019. As it is indicated in the paper being reviewed, the daily rainfall totals are important and new element for Saudi Arabia precipitation research. The publication delivers information on rainfall distribution over (across)  Saudi Arabia and its trends. Introduction chapter delivers the literature overview on rainfall variability an extreme events problem in Saudi Arabia. It introduces into the subject of the publication and formulates the aim.

Data and methods: Daily rainfall records were collected from 25 meteorological station, the lacking data were supplemented. Monthly data comes from two different sources (as gridded data used for climatological analysis. The data from both sources were compared. The analysis was performed at the annual, wet and dry seasons and monthly scale as well as for meteorological stations data at the daily scale. The analysis of data quality was performed but I cannot see any information about this in the results. In the paper, the extreme amounts are understood as daily maxima (even 1 mm) but also “a daily rainfall of ≥26 mm is identified as the threshold for an extreme event”. The definition of extreme event used in the paper should be written in the Methods instead of in the Results. I propose to put some more methodological information from the Results into the Methods. For instance, what for the percentile distribution was used and to explain that it can be used to predict the frequency of event for the next years.

The disadvantage of the paper is lack of discussion but some information on climate impact is inserted in the Results. The publication is long and deliver plenty of information that becomes less and less absorbed while reading the text.

The other remarks, I put according to the rows order:

Row 52: change the font

Row 61: rainy days (not rainy day)

Row 381: Figure 9. The legend is cut off. The same is for Figure 14.

Row 397: Figure 10. Could You overthink and change the titles of the table and Y axis, please. “Day/decade” - is it number of days per decade?

Rows 442-473: explain the term “rainfall return period” shortly in the Methods. Could it be an indicator for future prediction? If yes, it is worth to emphasize it in the text and conclusions. If the return period is 2.05 – it mean that we can expect such totals a little more than every two years? It the period is 43 – it mean that once per 43 years?

Rows: 474-519: “Contribution of extreme rainfall to the total rainfall”. The same as above. I advice to delete the table 14 and information concerning it.

Row 524: The word "record" has double meaning: as the highest value or as the register. It is better to write register, more clear for superficial readers.

Row 681-683: Put the year of edition in the proper place.

I recommend the article for publishing after considering the comments described above.

Author Response

Reviewer 3

Comments and Suggestions for Authors:

I read the article with interest. It is written clearly, in the proper order of information and is understandable to the reader. The analysis covers the period 1978-2019. As it is indicated in the paper being reviewed, the daily rainfall totals are important and new element for Saudi Arabia precipitation research. The publication delivers information on rainfall distribution over (across) Saudi Arabia and its trends. Introduction chapter delivers the literature overview on rainfall variability an extreme events problem in Saudi Arabia. It introduces into the subject of the publication and formulates the aim.

Response: Thanks for your encouraging comments and suggestion that improved the manuscript.

Data and methods: Daily rainfall records were collected from 25 meteorological station, the lacking data were supplemented. Monthly data comes from two different sources (as gridded data used for climatological analysis. The data from both sources were compared. The analysis was performed at the annual, wet and dry seasons and monthly scale as well as for meteorological stations data at the daily scale. The analysis of data quality was performed but I cannot see any information about this in the results. In the paper, the extreme amounts are understood as daily maxima (even 1 mm) but also “a daily rainfall of ≥26 mm is identified as the threshold for an extreme event”. The definition of extreme event used in the paper should be written in the Methods instead of in the Results. I propose to put some more methodological information from the Results into the Methods. For instance, what for the percentile distribution was used and to explain that it can be used to predict the frequency of event for the next years.

Response: Following the reviewer’s advice, the data and methodology section are revised and the definition of extreme event is given in the methodology part.

The disadvantage of the paper is lack of discussion but some information on climate impact is inserted in the Results. The publication is long and deliver plenty of information that becomes less and less absorbed while reading the text.

Response: I thank the reviewer for his comment. In the revised version, the text has been improved to deliver the message more clearly.

The other remarks, I put according to the rows order:

Row 52: change the font

Response: Correction made accordingly.

Row 61: rainy days (not rainy day)

Response: Following reviewer #2 comments, this sentence is removed from the text in the revised version.

Row 381: Figure 9. The legend is cut off. The same is for Figure 14.

Response: Thanks for this comment. The legends are now visible in the figures.

Row 397: Figure 10. Could You overthink and change the titles of the table and Y axis, please. “Day/decade” - is it number of days per decade?

Response: Followed the suggestion, it is the slope of the count time series at a specific daily rainfall threshold.

Rows 442-473: explain the term “rainfall return period” shortly in the Methods. Could it be an indicator for future prediction? If yes, it is worth to emphasize it in the text and conclusions. If the return period is 2.05 – it mean that we can expect such totals a little more than every two years? It the period is 43 – it mean that once per 43 years?

Response: Followed the reviewer’s suggestion. Rainfall Return Period is now defined in the Methodology section. The explanation is correct that if the period is 2.05 or 43 years means the amount is expected to return within such year.

Rows: 474-519: “Contribution of extreme rainfall to the total rainfall”. The same as above. I advise to delete the table 14 and information concerning it.

Response: Thanks for this comment. We do not have Table 14, and there are only 2 Tables in the manuscript. If it is Fig. 14, I think this figure is essential to define the threshold for extreme events that contribute to total rainfall.

Row 524: The word "record" has double meaning: as the highest value or as the register. It is better to write register, more clear for superficial readers.

Response: Followed the suggestion.

Row 681-683: Put the year of edition in the proper place.

Response: Followed the suggestion

I recommend the article for publishing after considering the comments described above.

Submission Date: 24 July 2020

Date of this review: 10 Aug 2020 09:33:42

Response: The author thanks the reviewers for their worthy comments. The comments/suggestions from the reviewer helped a lot in improving the quality of the paper.

Round 2

Reviewer 2 Report

2nd review of atmosphere-893402

It can be seen that the authors improved the manuscript based on the reviewers' comments. My recommendation is to Accept the paper. 

Please remove this sentence as it presents false information. 

L159 "There are also modified Mann-Kendall versions; however, the
modified version has almost no change for rainfall and its extremes [17]. "

Author Response

Manuscript # atmosphere-893402 entitled "Rainfall trends and extremes in Saudi Arabia in the recent decades" which you submitted to the Atmosphere, has been reviewed.

I am very happy to accept the paper after minor revision.

Reviewer 2 asked to remove the following sentence as it presents false information.

L159 "There are also modified Mann-Kendall versions; however, the modified version has almost no change for rainfall and its extremes [17].

Thank you for submitting your manuscript to the Atmosphere.

Response: Thank you very much for your comments and suggestion to improve the quality of the manuscript.

The sentence "There are also modified Mann-Kendall versions; however, the modified version has almost no change for rainfall, and its extremes [17] is removed from the revised version.